# SIX1 and EWS/FLI1 co-regulate an anti-metastatic gene network in Ewing Sarcoma

Connor J. Hughes [1,2,3], Kaiah M. Fields[3,4], Etienne P. Danis[3], Jessica Y. Hsu[3], Deepika Neelakantan[3,4,8], Melanie Y. Vincent[3,9], Annika L. Gustafson[1,3,4], Michael J. Oliphant[3,5,10], Varsha Sreekanth[3], Vadym Zaberezhnyy[6], James C. Costello [1,2,3], Paul Jedlicka[1,7] & Heide L. Ford [1,2,3,4] ✉

Ewing sarcoma (ES), which is characterized by the presence of oncogenic fusion proteins such as EWS/FLI1, is an aggressive pediatric malignancy with a high rate of early dissemination and poor outcome after distant spread. Here we demonstrate that the SIX1 homeoprotein, which *enhances* metastasis in most tumor types, suppresses ES metastasis by co-regulating EWS/FLI1 target genes. Like EWS/FLI1, SIX1 promotes cell growth/transformation, yet dramatically inhibits migration and invasion, as well as metastasis in vivo. We show that EWS/FLI1 promotes SIX1 protein expression, and that the two proteins share genome-wide binding profiles and transcriptional regulatory targets, including many metastasis-associated genes such as integrins, which they co-regulate. We further show that SIX1 downregulation of integrins is critical to its ability to inhibit invasion, a key characteristic of metastatic cells. These data demonstrate an unexpected anti-metastatic function for SIX1, through coordinate gene regulation with the key oncoprotein in ES, EWS/FLI1.

Ewing sarcoma (ES) is the second most common pediatric malignancy of the bone and soft tissue[1], predominantly affecting adolescents and young adults. ES tumors are highly aggressive; 25% of cases display evidence of metastatic disease[1], and the 5-year survival rate is below 30% for these patients[2]. Thus, there is an urgent need to understand the mechanisms regulating metastasis in ES.

Most ES tumors harbor an (11;22)(q24;q12) chromosomal translocation[3], leading to the expression of the chimeric transcription factor EWS/FLI1, which unites the N-terminus of the EWSR1 transcriptional activator with the C-terminus of the ETS family transcription factor (TF) and DNA-binding protein, FLI1[4]. Recent studies demonstrate that ES cells express variable levels of EWS/FLI1, and intriguingly, high EWS/FLI1 and low EWS/FLI1 expression are associated with distinct phenotypic states[5]. High EWS/FLI1-expressing cells exhibit increased proliferative indices and strong cell-cell interactions[5], and represent the majority of cells in an ES tumor. In contrast, EWS/FLI1-low cells are less abundant and less proliferative than their EWS/FLI1-high counterparts, but are more migratory and invasive and promote distant dissemination of ES cells[5,6]. In line with this observation, expression of EWS/FLI1 results in the loss of focal adhesions and actin stress fibers[7], as well as the repression of multiple integrins[5–7], decreasing the ability of ES cells to adhere to the extracellular matrix (ECM) and to migrate/invade[7].

[1]Medical Scientist Training Program, University of Colorado Anschutz Medical Campus, Aurora, CO 80045, USA. [2]Pharmacology Program, University of Colorado Anschutz Medical Campus, Aurora, CO 80045, USA. [3]Department of Pharmacology, University of Colorado Anschutz Medical Campus, 12800 East 19th Avenue, Aurora, CO 80045, USA. [4]Molecular Biology Program, University of Colorado Anschutz Medical Campus, Aurora, CO 80045, USA. [5]Integrative Physiology Program, University of Colorado Anschutz Medical Campus, Aurora, CO 80045, USA. [6]Department of Biochemistry and Molecular Genetics, University of Colorado Anschutz Medical Campus, Aurora, CO 80045, USA. [7]Department of Pathology, University of Colorado Anschutz Medical Campus, Aurora, CO 80045, USA. [8]Present address: OU Health Stephenson Cancer Center, Oklahoma City, OK 73104, USA. [9]Present address: Vigeo Therapeutics, 85 Bolton St, Cambridge, MA 02140, USA. [10]Present address: Department of Cell Biology, Harvard Medical School, Boston, MA 02115, USA. ✉e-mail: heide.ford@cuanschutz.edu

The mammalian gene *Six1* is a vertebrate homolog to the *sine oculis (so)* gene in *Drosophila*, which plays a pivotal role in *Drosophila* eye development[8]. Mammalian Six family members (Six1-6) play similar roles during mammalian organogenesis[9] and *SIX1* expression is typically reduced in normal adult tissue[10]. However, re-expression of *SIX1* has been demonstrated in a host of distinct tumor types, including breast cancer[11], rhabdomyosarcoma (RMS)[12–14], and hepatocellular carcinoma[15] amongst others[16–20]. SIX1 in cancer cells can drive transcriptional activation of pro-tumorigenic[13] and pro-metastatic[11,14,15] gene sets and promote multiple "hallmarks of cancer" including enhanced proliferation[17,21], resistance to cell death[19,22], and enhanced invasiveness/metastatic potential[11,13,23]. As a consequence of these tumor-promotional functions, SIX1 overexpression is almost uniformly considered to be a poor prognostic indicator across many diverse tumor types[16,19,20,23]. With the exception of a small subset of sarcomas, including RMS[12,13] and osteosarcoma[20] (OS), the functions of SIX1 have primarily been studied in the context of tumors of epithelial origin. As many of the pro-migratory/pro-invasive phenotypes driven by SIX1 overexpression are linked to the promotion of an epithelial-to-mesenchymal transition (EMT)[11,18], whether these same phenotypes are driven by SIX1 in tumor types of mesenchymal origin is largely unknown.

In this report, we demonstrate that the functions and transcriptional regulatory targets of SIX1 in ES mimic that of the key oncogenic EWS/FLI1 fusion protein. While SIX1 promotes anchorage-independent growth of ES cells, it suppresses their migration and invasion and inhibits late-stage metastatic outgrowth in mice. We demonstrate that EWS/FLI1 promotes SIX1 protein expression, and SIX1 binds directly to the promoters of, and inhibits, the expression of multiple integrins, and suppression of integrin activity is sufficient to inhibit in vitro invasive phenotypes seen in SIX1 knockdown (KD) cells. Intriguingly, combined CUT&RUN and transcriptomic analyses demonstrate profound genome-wide binding overlap and extensive co-regulated genes between SIX1 and EWS/FLI1, many of which are involved in regulating metastatic processes. Importantly, the relative expression of SIX1 and EWS/FLI1 can lead to distinct patterns of integrin regulation, demonstrating that SIX1 and EWS/FLI1 are together orchestrating a complex anti-metastatic gene network in ES. This anti-metastatic function for SIX1 is in contrast to previous reports demonstrating pro-metastatic functions for SIX1 in other tumor types, underscoring the importance of genetic background when studying developmental regulators in cancer. As we begin to assess the anti-cancer utility of SIX1 complex-targeted therapies, currently under development by our and other laboratories[10,24,25], it is vital to understand tumor-specific functions of SIX1 in order to guide safe and effective use of such therapies.

## Results

### SIX1 expression is increased in ES relative to mesenchymal stem cells

While SIX1 is a known driver of EMT and metastasis in the context of carcinomas[10,11,15,18] whether it similarly promotes metastasis in tumors of mesenchymal origin is far less understood. Previous analysis of the National Cancer Institute (NCI) Oncogenomics Sarcoma Gene Expression dataset demonstrated that SIX1 mRNA is overexpressed in ES tumors relative to normal tissue (average expression across multiple adult tissues)[12], but is not nearly as upregulated as it is in other sarcomas where it is known to promote tumor aggressiveness[12,20,22,26]. This finding prompted us to further test whether SIX1 may play a functionally distinct role in ES when compared to other sarcomas.

To identify cell lines that could be used for functional studies, we first assessed SIX1 mRNA and protein levels in a panel of ES cell lines and a candidate cell of origin for ES, human mesenchymal stem cells (hMSCs)[7]. Only the A673 cell line had increased SIX1 mRNA compared to hMSCs (Fig. 1a), whereas all examined ES cell lines displayed increased levels of SIX1 protein (Fig. 1b), suggesting that SIX1 is primarily controlled post-transcriptionally in ES. The commonly-studied

ES cell lines A673 and EWS-502 displayed the highest SIX1 mRNA and/or protein levels across this panel (Fig. 1a, b), and thus we selected these two lines for further investigation.

### Knockdown (KD) of SIX1 in ES inhibits anchorage-independent growth

To determine the role of SIX1 in ES, we performed shRNA-mediated knockdown (KD) of SIX1 with two different guide shRNA sequences in both A673 and EWS-502 cells, which was confirmed at the mRNA (Fig. 1c, d) and protein levels (Fig. 1e, f). As SIX1 is a known cell cycle regulator that drives proliferation in numerous tumors[17,22,26], we examined the overall growth rates of SCR and SIX1 KD cells in the A673 and EWS-502 cell lines using an IncuCyte growth assay. Surprisingly, SIX1 KD had no impact on the growth rate of A673 cells (Fig. 1g). However, KD of SIX1 in the EWS-502 cells led to a statistically significant reduction in growth over time (Fig. 1h). A673 cells harbor a BRAF$^{V600E}$ mutation and are also known to have constitutive activation of PI3K[27], and it has previously been shown that even KD of EWS/FLI1, the main driver of ES oncogenesis, does not inhibit the growth of A673 cells in adherent conditions, but does suppress anchorage-independent growth[7]. Thus, we tested whether anchorage-independent growth is affected by SIX1 loss. Indeed, SIX1 KD led to a significant reduction in colony formation in soft agar in both the A673 (Fig. 1i, j) and EWS-502 (Fig. 1k, l) cells, demonstrating that like EWS/FLI1, SIX1 promotes anchorage-independent growth of ES.

### SIX1 inhibits migration and invasion of ES cells

Because SIX1 is causally linked to metastasis in a host of tumor types[11,13,15,28–30], we investigated whether SIX1 may also regulate in vitro measures associated with increased metastatic potential, including transwell migration and ECM-invasion. To our surprise, SIX1 KD consistently led to significantly *enhanced* transwell migration in both the A673 (Fig. 2a, b) and EWS-502 cell lines (Fig. 2c, d). These results were additionally confirmed using stable SIX1 KD clones in the A673 cell line (Supplementary Fig. 1a) as well as with transient SIX1 KD in the EWS-502 cell line (Supplementary Fig. 1b), both which exhibited increased migration with SIX1 KD using scratch wound assays (Supplementary Fig. 1c–f). SIX1 KD also enhanced in vitro Matrigel invasion in both A673 pooled (Fig. 2e, f) and clonal (Supplementary Fig. 1g, h) KD populations and in EWS-502 pooled SIX1 KD (Fig. 2g, h) cells. As cell growth was unaffected or reduced in the ES lines (Fig. 1g, h) upon SIX1 KD, these results cannot be explained by SIX1 KD enhancing cell growth, and therefore represent true differences in their migratory/invasive potential in response to SIX1 loss.

### SIX1 inhibits metastasis in a murine late-stage ES metastasis model

SIX1 promotes metastasis in numerous tumor contexts[11,13,15,28,29]. However, the above data strongly suggest that it may, in stark contrast, be anti-metastatic in ES. Thus, to examine the role of SIX1 in metastasis, male NSG mice were injected via tail vein with firefly luciferase-tagged A673 SCR or SIX1 KD cells and were imaged weekly to determine the extent of metastatic outgrowth. We prioritized male mice in this study as males are disproportionately represented in ES incidence (over 60% of cases) and have significantly worse outcomes as compared to females[31]. Mice injected with SIX1 KD cells developed metastases significantly more quickly than mice injected with SCR cells (Fig. 3a, b), and had worse overall survival (Fig. 3c). Eventually, all mice injected with A673 cells developed metastases, demonstrating that higher SIX1 expression delays, but may not fully prevent, the development of metastasis of ES cells (Fig. 3c).

To rule out the possibility that the enhanced outgrowth observed with SIX1 KD after tail-vein injection was not a product of a tumorigenic or pro-proliferative phenotype of SIX1 KD cells in vivo, we performed subcutaneous flank injections of A673 SCR or SIX1 KD cells in NSG

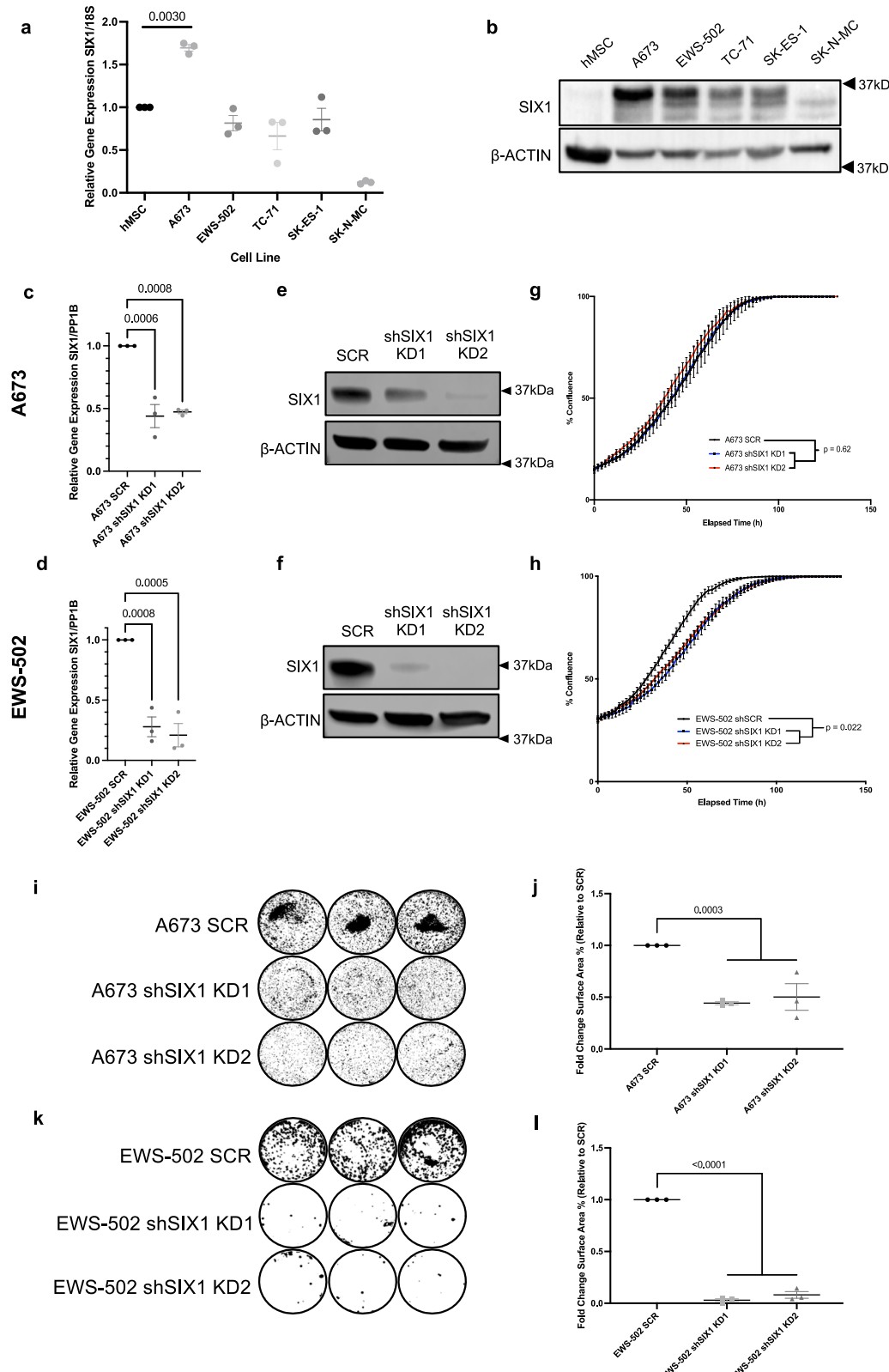

mice. In this subcutaneous model, SIX1 KD tumors had significantly decreased growth when compared to A673 SCR tumors (Supplementary Fig. 2), in line with the anchorage-independent growth phenotype seen in vitro. Given the fact that SIX1 loss inhibits growth in vivo, but enhances metastases, our findings suggest that SIX1 is directly suppressing late-stage metastasis in ES, rather than suppressing proliferation at the secondary site.

Because SIX1-mediated inhibition of metastasis contrasts with pro-metastatic functions observed in other tumor contexts[11,13,18,28], we asked whether this effect is generalizable across multiple models of ES. To this end, we injected male NSG mice via tail vein with EWS-502 SCR or SIX1 KD cells, which were untagged, as luciferase-tagging decreased the overall viability of these cells. At 33 days post-injection, mice were sacrificed and necropsies were performed to look for evidence of

**Fig. 1 | SIX1 is overexpressed in Ewing sarcoma (ES) compared to mesenchymal stem cells, and mediates growth in soft agar. a** qRT-PCR assay of SIX1 gene expression comparing human mesenchymal stem cells (hMSCs) to a panel of ES cell lines. Mean ± SEM of $n = 3$ independent experimental replicates shown. Statistical analysis was performed using unpaired two-tailed Welch's T-test. **b** Western blot analysis of SIX1 protein expression comparing the same cell lines as (**a**). qRT-PCR expression comparing SIX1 levels in **c** A673 and **d** EWS-502 SCR and SIX1 KD cell lines. Mean ± SEM of $n = 3$ independent experimental replicates shown. Statistical analysis was performed using one-tailed ANOVA with post-hoc Dunnett's multiple comparisons test. Protein expression comparing SIX1 levels in **e** A673 and **f** EWS-502 SCR and SIX1 KD cell lines. Incucyte growth assays comparing SCR and SIX1 KD cells in the **g** A673 and **h** EWS-502 cell lines. Statistical analysis for each incucyte assay was performed by fitting data to a longitudinal mixed-effects model. **i** Representative images from soft agar growth assays in the A673 cell line. **j** Quantification of $n = 3$ independent experimental replicates of the assay shown in (**i**). Mean ± SEM shown. Statistical analysis comparing SCR and combined SIX1 KD replicates performed using unpaired two-tailed Welch's T-test. **k** Representative images from soft agar growth assays in the EWS-502 cell line. **l** Quantification of $n = 3$ independent experimental replicates of the assay shown in (**k**). Mean ± SEM shown. Statistical analysis comparing SCR and combined SIX1 KD replicates performed using unpaired two-tailed Welch's T-test. Each experimental replicate consists of $n = 3$ technical replicates (averaged). All Western blots are representative results of three independent replicate experiments. Source data are provided as a Source Data file.

metastatic outgrowth. With the exception of one supra-hilar mass, metastatic lesions were localized to the liver, and by gross examination, all metastases that arose were observed in mice injected with EWS-502 SIX1 KD cells (4/9 mice). To confirm this observation, we fixed liver sections and performed hematoxylin and eosin (H&E) staining. Histologic examination of the sections confirmed that metastatic lesions were only detected in mice injected with EWS-502 SIX1 KD cells (4/9), and that no metastases were detected in mice injected with SCR cells (0/5) (Fig. 3d, e). Taken together, these data demonstrate that in the context of ES, SIX1 dramatically inhibits metastasis.

## SIX1 regulates a mixed EMT phenotype in ES

The results above demonstrate that in the context of ES, the impact of SIX1 expression on metastasis is opposite to that in most other described tumor types. Thus, to better understand how SIX1 inhibits migration, invasion, and metastasis in the context of ES, we performed bulk RNA-sequencing (RNA-seq) comparing A673 and EWS-502 SCR cells to respective SIX1 KD populations. GSEA analysis of our RNA-seq data demonstrated that the Hallmark Epithelial-to-Mesenchymal Transition pathway was one of the top negatively enriched gene sets upon SIX1 KD (Fig. 4a, b) in both the A673 and EWS-502 cell lines (Fig. 4c, d).

Because our RNA-seq data suggest that SIX1 is largely promoting EMT gene expression in both ES lines, in a similar manner to its regulation in other tumor types[10,11,18], and yet we observed pro-metastatic cell behaviors associated with SIX1 loss, we reasoned it is unlikely that SIX1 is regulating a bona fide EMT phenotype in ES. We thus performed qRT-PCR on a panel of EMT-regulated genes and EMT-TFs[32], many of which are known to be regulated by SIX1 in other tumor types[10,11,18], in the A673 and EWS-502 systems. SIX1 KD led to a significant reduction in the expression of CDH1, CDH2, SNAI1, TWIST1, VIM, and FN1 in the A673 system (Supplementary Fig. 3a–f). CDH1, CDH2, SNAI1, and VIM were similarly reduced with SIX1 KD in the EWS-502 system, but we observed mixed effects on TWIST1 and FN1 (Supplementary Fig. 3g–l). The fact that SIX1 loss leads to the downregulation of both epithelial (CDH1) and mesenchymal (CDH2) genes, and also differentially effects EMT-TFs, suggests a nuanced, and complex, role for SIX1 in altering EMT parameters in the context of ES.

Given this mixed-EMT transcriptional signature, we asked whether SIX1 expression in ES promotes phenotypes associated with a canonical EMT, such as apoptotic- or chemo-resistance[33]. Propidium Iodide (PI) staining and flow cytometric analysis to look for differences in late-stage apoptosis between SCR and SIX1 KD ES cells also gave mixed results, with A673 cells displaying no significant difference in apoptosis upon SIX1 KD (Supplementary Fig. 4a), whereas EWS-502 cells showed a reduction in apoptosis with SIX1 KD (Supplementary Fig. 4b). We additionally performed in vitro chemo-resistance assays to determine the IC50 of SCR and SIX1 KD cells to a pair of standard-of-care chemotherapeutic agents used clinically for ES, Vincristine and Actinomycin D[34]. We did not observe any change in the IC50 of Vincristine with SIX1 KD in either system (Supplementary Fig. 4c, d). However, we did observe a significant reduction in the IC50 of Actinomycin D with SIX1 KD in both systems (Supplementary Fig. 4e, f), demonstrating that SIX1 promotes

class-specific, rather than generalized, chemo-resistance. These data demonstrate that SIX1 expression and loss may regulate specific epithelial or mesenchymal cellular features, but that SIX1 does not regulate a bona fide EMT in the context of ES.

## SIX1 inhibits integrin expression and signaling in ES cells

To gain additional insight into how SIX1 represses migration and invasion in the context of ES in a manner not dependent on a traditional EMT, we performed over-representation analysis (ORA) on both RNA-seq datasets to look for Gene Ontology (GO) gene sets that are significantly altered with SIX1 KD. A673 and EWS-502 SIX1 KD lines exhibited significant changes in the regulation of genes involved in cell motility and adhesion (Fig. 4e, f); pathways known to be critical for metastasis.

To further explore how SIX1 may regulate motility and adhesion in ES, we performed KEGG Topology analysis on our RNA-seq datasets. This analysis demonstrated that the ECM-Receptor Interaction pathway is highly regulated by SIX1 in both the A673 and EWS-502 cell lines (Supplementary Fig. 5a, b). Within this KEGG pathway, we observed strong enrichment for numerous integrin mRNAs in SIX1 KD cells relative to SCR cells in both systems (Supplementary Fig. 6a, b). We confirmed these findings in both cell lines on independent RNA samples for ITGA1, 2, 4, 5, and 6 as well as ITGB1 by qRT-PCR (Fig. 5a, b), but not for ITGA3 (Supplementary Fig. 7a, b). As integrins involved in adhesion and signaling are membrane-localized, we performed Western blot analysis on extracted membrane-bound protein fractions from A673 and EWS-502 SCR and SIX1 KD cells, which demonstrated increased levels of membrane-localized integrins α1, α2, α4, α5, α6, and β1 in response to SIX1 KD (Fig. 5c, d).

In addition to their direct roles in adhesion, integrin complexes can activate intracellular signaling cascades largely through activation of focal adhesion kinase (FAK) and c-Src kinase[35] via phosphorylation of Y397[35] and Y416[35,36] on FAK and SRC, respectively. Activation of the FAK-SRC complex is vital for regulating the turnover of focal adhesions necessary for coordinated tumor cell motility and mesenchymal invasion[36,37]. Thus, we performed Western blotting analysis to look for alterations in the expression level of phospho-FAK (Y397) and phospho-SRC (Y416) between SCR and SIX1 KD cells, and observed that the activated phospho-form of both FAK and SRC were strongly enriched in SIX1 KD cells in both systems (Fig. 5e, f). These data demonstrate that not only does SIX1 KD induce surface integrin expression, but it also promotes intracellular integrin-mediated signaling which is important for cellular motility and invasiveness.

## SIX1 KD leads to enhanced ECM adhesion and inhibition of integrin activity abrogates the pro-invasive phenotype observed with SIX1 KD

Integrins facilitate cancer cell adhesion to the ECM and invasion through interstitial tissue, and increased integrin expression is associated with tumor progression in numerous tumor types, including ES[6,38,39]. Thus, we examined whether SIX1 repression of integrins is

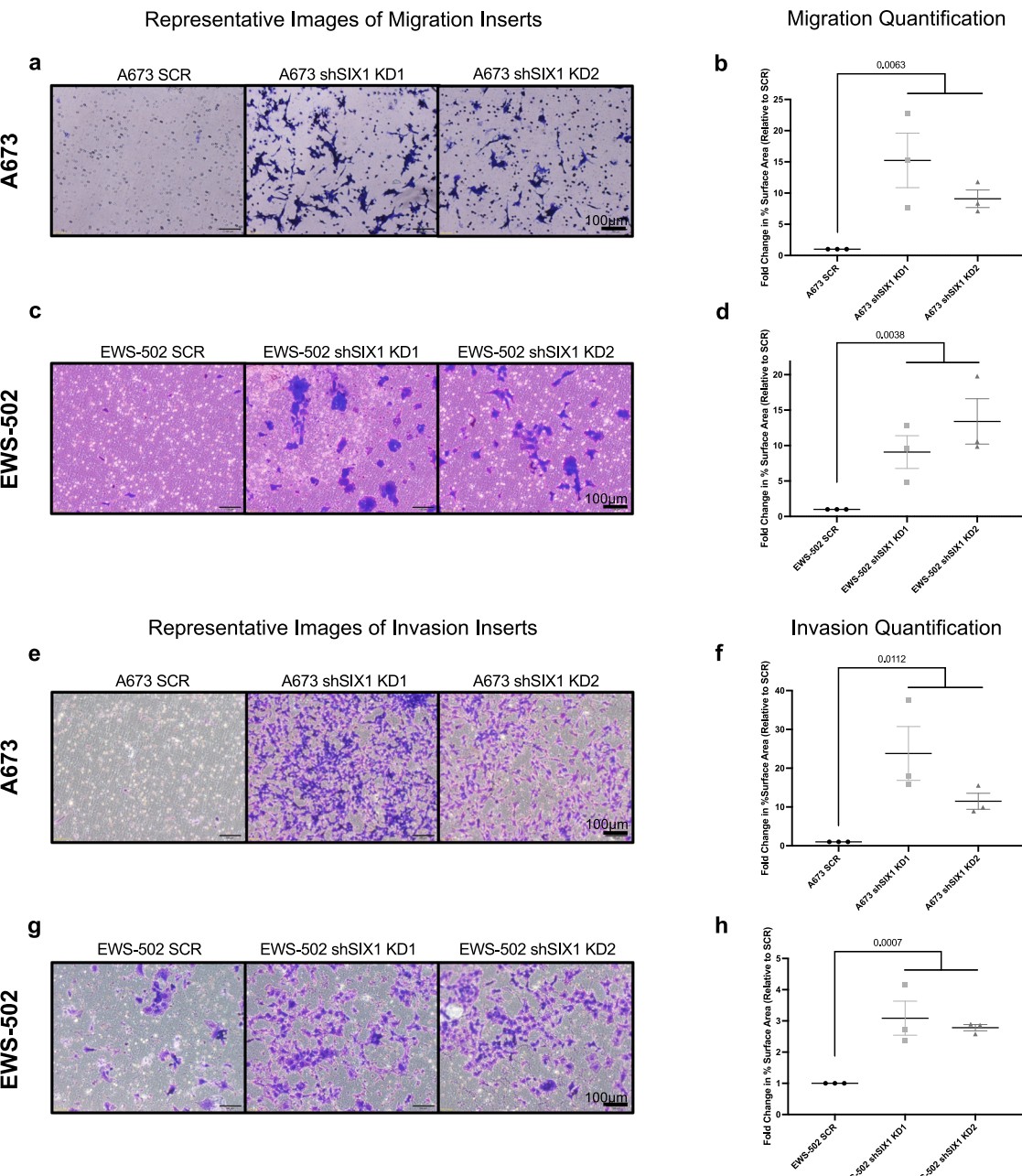

**Fig. 2 | SIX1 KD enhances migration and invasion. a** Representative images from A673 SCR and SIX1 KD transwell migration inserts after 24 h. **b** Quantification of $n = 3$ independent experimental replicates. Mean ± SEM of fold change (relative to SCR) shown. **c** Representative images from EWS-502 SCR and SIX1 KD transwell migration inserts after 40 h. **d** Quantification of $n = 3$ independent experimental replicates. Mean ± SEM of fold change (relative to SCR) shown. **e** Representative images from A673 SCR and SIX1 KD transwell invasion inserts after 72 h. **f** Quantification of $n = 3$ independent experimental replicates. Mean ± SEM of fold change (relative to SCR) shown. **g** Representative images from EWS-502 SCR and SIX1 KD transwell invasion inserts after 96 h. **h** Quantification of $n = 3$ independent experimental replicates. Mean ± SEM of fold change (relative to SCR) shown. Statistical analysis comparing SCR and combined SIX1 KD replicates performed using unpaired two-tailed Welch's T-test. Each experimental replicate consists of $n = 3$ technical replicates (averaged). Scale bars = 100 μm. Source data are provided as a Source Data file.

critical for its ability to inhibit metastasis-associated phenotypes in ES cells. Because different heterodimeric integrins bind different ECM components, adhesion assays were performed with collagen IV, fibronectin, and laminin. KD of SIX1 enhanced the binding of A673 and EWS-502 cells to all tested matrices (Fig. 6a–f), suggesting that the observed broad increases in integrin expression with SIX1 KD result in functional increases in ECM adhesion.

To determine whether SIX1-mediated alterations in integrin expression contribute to the observed differences in the invasive potential of SIX1 KD vs. SCR cells, we performed transwell invasion assays utilizing neutralizing antibodies for a panel of integrins to block their activity. Neutralization of β1, α1β1, α2β1, and α4β1 independently led to reductions in the invasive potential of A673 SIX1 KD cells (Fig. 6g, h) and EWS-502 SIX1 KD cells (Fig. 6i, j), while having little, if any, impact on the relatively low invasive potential of SCR cells, demonstrating that SIX1 KD-induced integrin expression is vital for the invasive phenotype of SIX1 KD cells. However, whether the regulation of integrins is solely responsible for the pro-metastatic phenotype of these cells, or whether other targets of SIX1 are critical, remains to be determined.

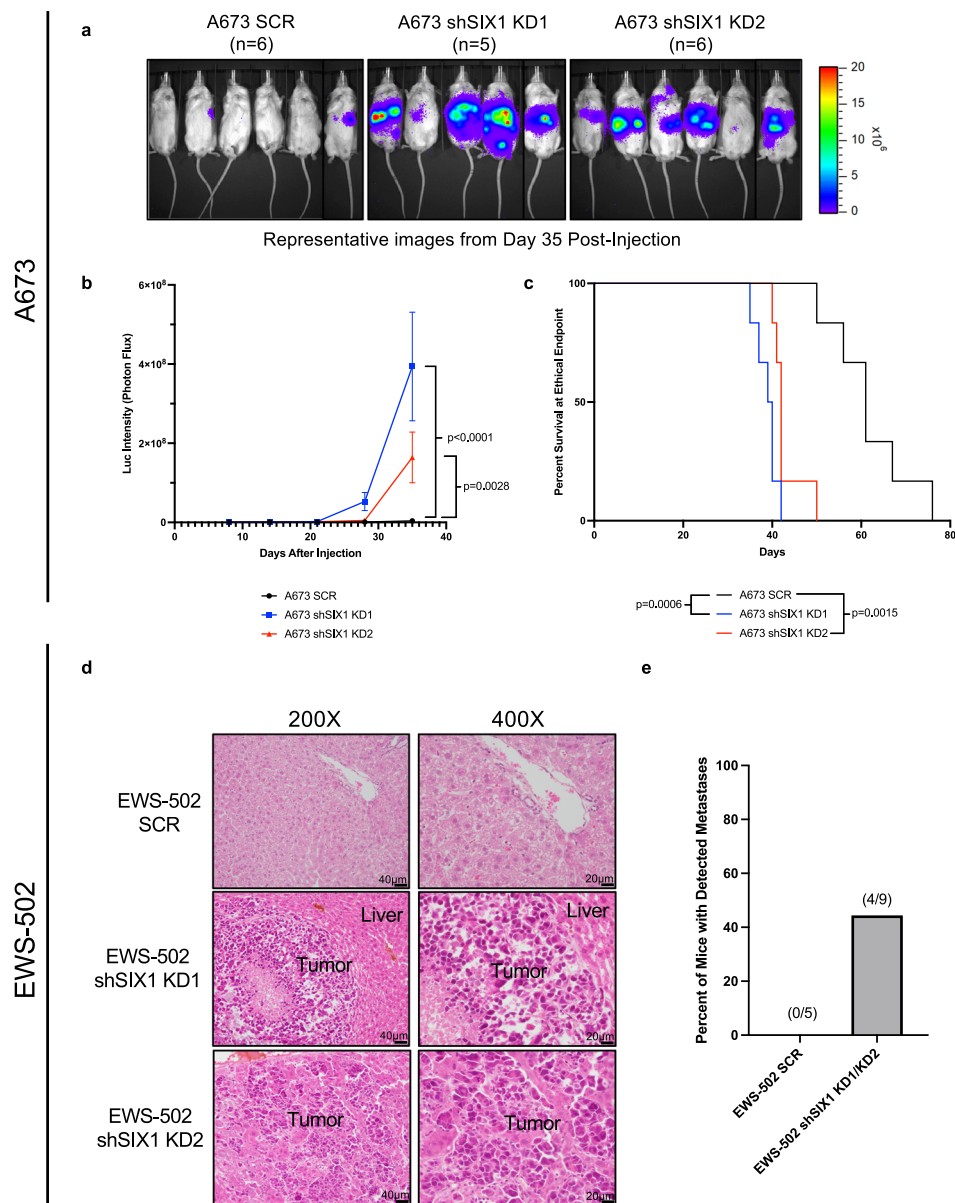

**Fig. 3 | SIX1 KD increases late-stage metastasis and decreases survival.**
**a** Representative IVIS images of mice injected with A673 SCR, A673 shSIX1 KD1, or A673 shSIX1 KD2 cells via tail vein on day 35 post-injection ($n = 6$ mice/group, $n = 5$ mice for shSIX1 KD1 at Day 35). **b** Quantification of body-wide luciferase intensity after injection of cells described in (**a**). Mean ± SEM shown. Statistical analysis was performed using two-way ANOVA with post-hoc Dunnett's multiple comparisons test ($n = 6$ mice/group, $n = 5$ mice for shSIX1 KD1 at Day 35). **c** Survival of mice after tail vein injection of cells described in (**a**). ($n = 6$ mice/group). Statistical analysis was performed using the log-rank test. **d** 200X and 400X magnification images of representative H&E-stained liver sections from mice injected with EWS-502 SCR ($n = 5$ mice) or shSIX1 KD ($n = 9$ mice) cells via tail vein. **e** Percentage of mice that had pathologic metastases after tail vein injection with either EWS-502 SCR or shSIX1 KD1/KD2 cells. Scale bars: 200X; 40 μm, 400X; 20 μm. Source data are provided as a Source Data file.

## EWS/FLI1 regulates SIX1 protein levels

Because SIX1 has the opposite effect on ES metastasis than it does in all previously studied tumor types[11,13–16,18,28–30], we hypothesized that this anti-metastatic effect may be due to the concomitant expression of the transgene EWS/FLI1, which regulates many phenotypes in a similar manner to what we have observed with SIX1, including promoting anchorage-independent growth (Fig. 1i–l) while repressing metastatic features (Fig. 2a–h)[6,7]. Thus, we first asked whether SIX1 may regulate EWS-FLI1 levels, or vice-versa. SIX1 KD in A673 or in EWS-502 cells did not lead to any appreciable reduction in the expression of EWS/FLI1 (Supplementary Fig. 8a, b). Using a doxycycline-inducible EWS/FLI1 KD system in A673 cells[40], we observed no consistent change in SIX1 mRNA expression with EWS/

FLI1 KD (Supplementary Fig. 8c, d), but did observe a dose-dependent reduction in SIX1 protein with EWS/FLI1 KD (Supplementary Fig. 8e), demonstrating that EWS/FLI1 regulates SIX1 protein levels in a non-transcriptional manner. To rule out non-specific effects of doxycycline treatment, we treated A673 SCR cells with the same dose of doxycycline (2 μg/ml) for 48 hours and observed no change in SIX1 protein levels (Supplementary Fig. 8f), demonstrating that this reduction in SIX1 is specific to EWS/FLI1 loss.

To further confirm these findings, we utilized shRNA-mediated KD of FLI1 in both the A673 and EWS-502 systems to look at effects on SIX1 expression. In line with the doxycycline-inducible EWS/FLI1 KD model, shRNA-mediated KD of FLI1 did not lead to consistent changes in SIX1 mRNA in either the A673 (Fig. 7a, b) or EWS-502

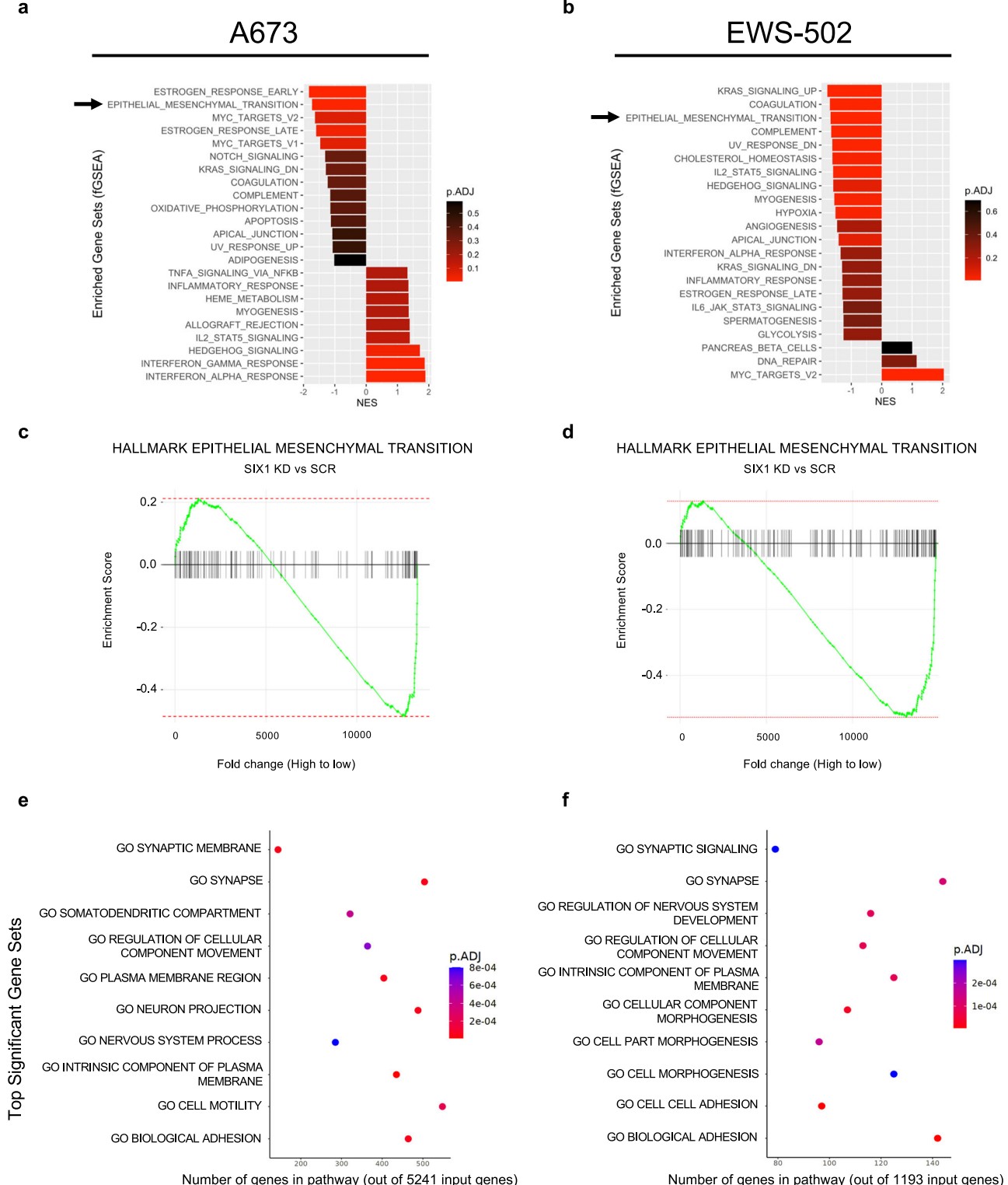

**Fig. 4 | SIX1 regulates EMT-, adhesion-, and motility-related gene sets. a** Top enriched Hallmark gene sets by Gene Set Enrichment Analysis (GSEA) upon SIX1 KD in the A673 cell line. **b** Top enriched Hallmark gene sets by GSEA upon SIX1 KD in the EWS-502 cell line. *P*-values for **a** and **b** are estimated from an adaptive multi-level split Monte-Carlo scheme with Benjamini-Hochberg multiple testing correction. **c** GSEA plot for Hallmark Epithelial-Mesenchymal Transition gene set comparing SIX1 KD to SCR cells in A673 cell line. **d** GSEA plot for Hallmark Epithelial-Mesenchymal Transition gene set comparing SIX1 KD to SCR cells in EWS-502 cell line. **e** Enriched Gene Ontology (GO) gene sets by over-representation analysis (ORA) comparing SIX1 KD to SCR cells in the A673 cell line. **f** Enriched GO gene sets by ORA comparing SIX1 KD to SCR cells in EWS-502 cell line. Statistics for **e** and **f** were calculated using a hypergeometric test (one-tailed Fisher's exact test). Source data are provided as a Source Data file and at GEO GSE215416.

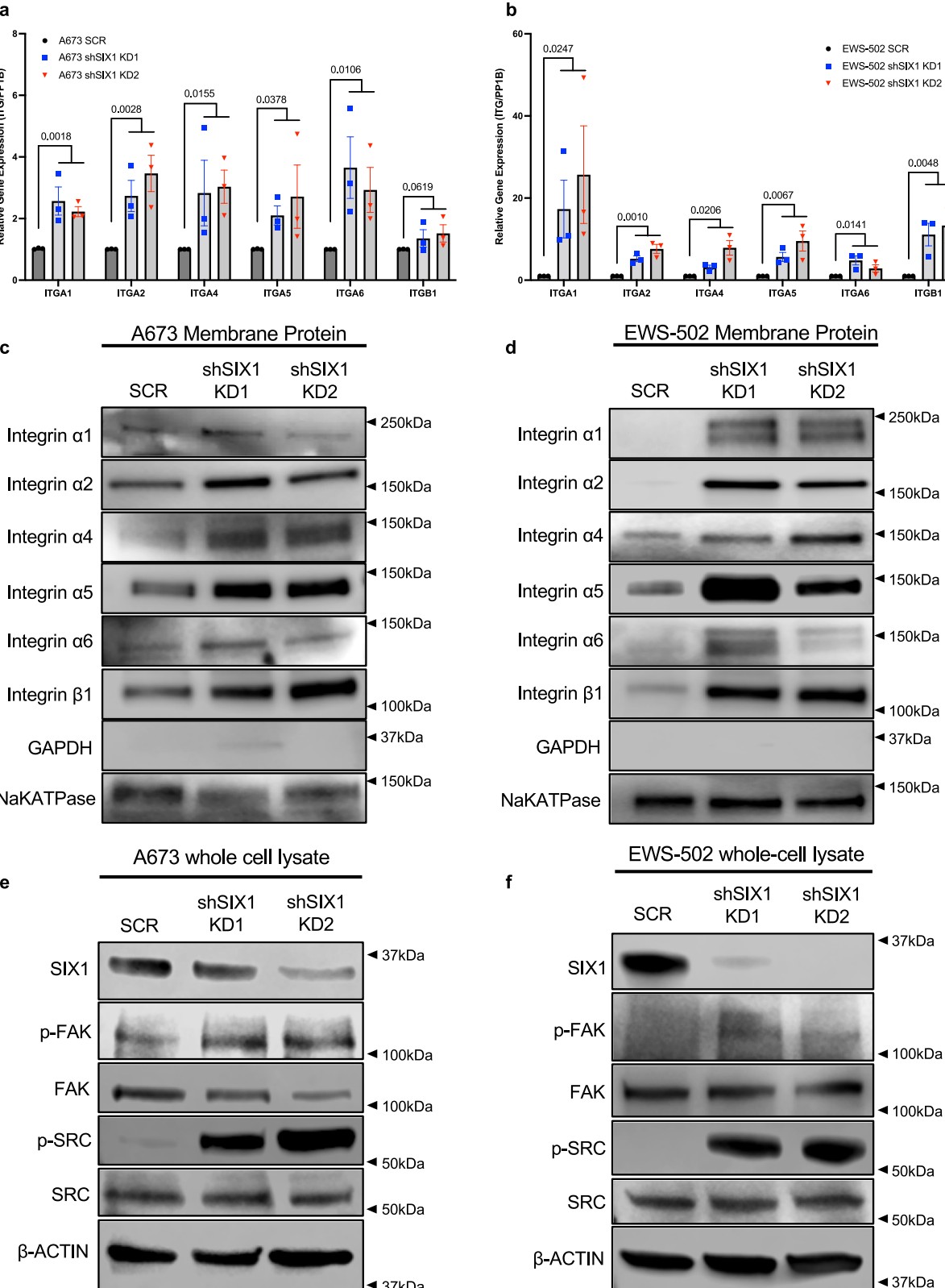

**Fig. 5 | SIX1 KD increases membrane expression and downstream activation of integrins.** qRT-PCR assays showing relative gene expression of ITGA1, ITGA2, ITGA4, ITGA5, ITGA6, and ITGB1 between SCR and SIX1 KD cells for the **a** A673 and **b** EWS-502 cell lines. Mean ± SEM of $n = 3$ independent experimental replicates shown. Each experimental replicate consists of $n = 3$ technical replicates (averaged). Statistical analysis was performed between SCR and combined SIX1 KD replicates per gene using an unpaired two-tailed Welch's T-test. Western blot analysis of membrane-specific expression of the aforementioned integrins between

SCR and SIX1 KD cells in **c** A673 and **d** EWS-502 cell lines. Representative GAPDH is shown as a membrane purity control and representative NaKATPase as a membrane protein loading control. Western blot analysis of activated p-FAK (Y379) and p-SRC (Y416) in SCR and SIX1 KD cells in the **e** A673 and **f** EWS-502 cell lines. Representative SIX1 and β-Actin blots are shown. Representative Western blots are shown for one of three independent experiments. Source data are provided as a Source Data file.

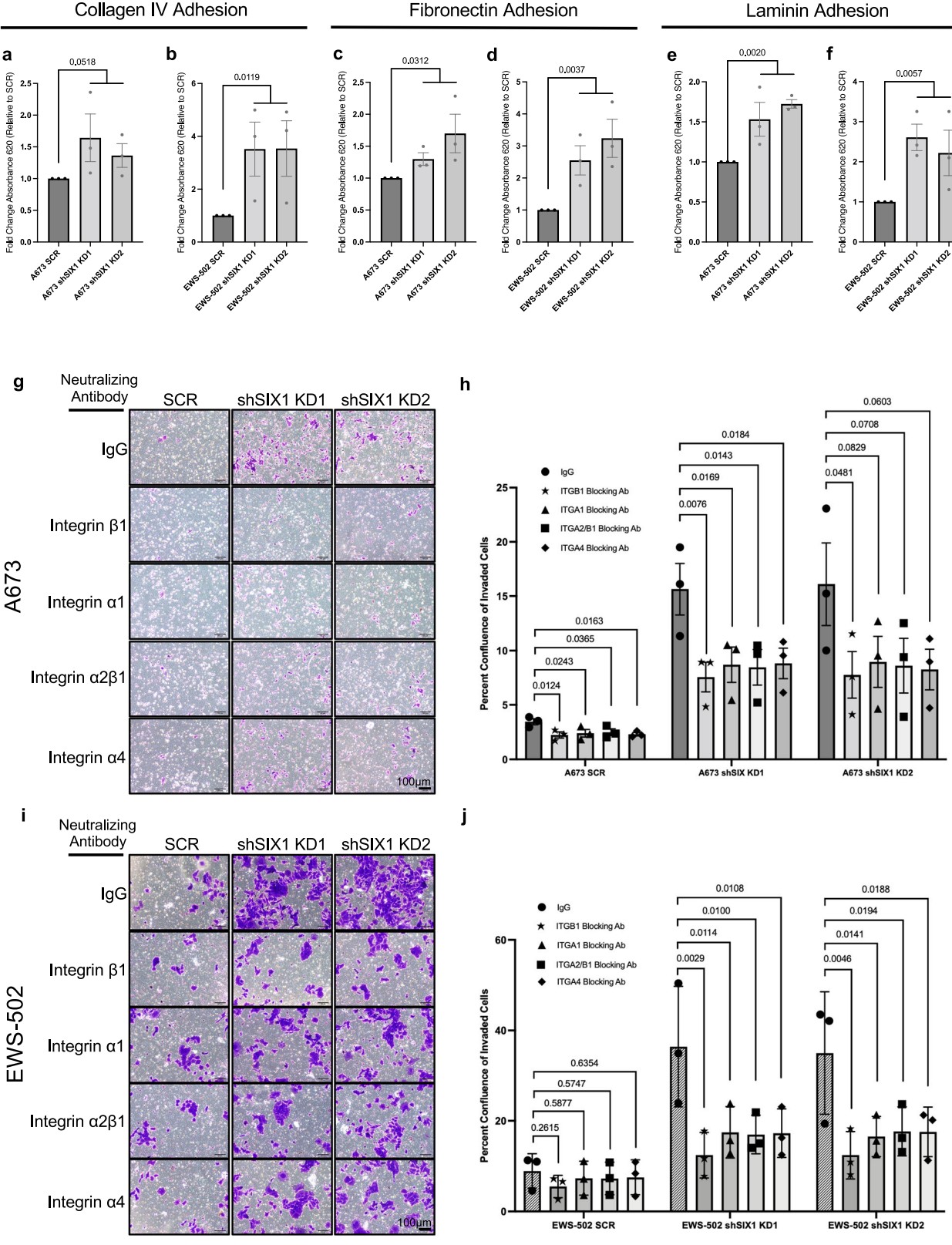

(Fig. 7c, d) systems, but did lead to marked reduction in SIX1 protein levels (Fig. 7e, f), demonstrating that EWS/FLI1 is a strong regulator of SIX1 protein expression in ES. Taken together, our data suggest that EWS/FLI1 can stabilize SIX1 in ES cells, which provides a potential explanation for the enrichment of SIX1 protein in ES cell lines relative to hMSCs (Fig. 1b).

## SIX1 and EWS/FLI1 bind to the same integrin promoters

The data above demonstrate that EWS/FLI1 is capable of regulating SIX1 protein expression and that SIX1 KD phenocopies EWS/FLI1-loss. Thus, we asked whether SIX1 and EWS/FLI1 may share overlapping regulatory targets. EWS/FLI1 has previously been shown to influence the expression of a number of integrins[6,7,38]. Given the observed

**Fig. 6 | SIX1 KD increases adhesion and inhibition of integrins reverses increases in invasion observed with SIX1 KD.** Quantification of adhesion as measured by crystal violet staining intensity for SCR and SIX1 KD cells to collagen IV- (**a**, **b**), fibronectin- (**c**, **d**), and laminin- (**e**, **f**) coated plates in the A673 and EWS-502 cell lines, respectively. Mean ± SEM for $n = 3$ independent experimental replicates shown for all adhesion assays. Each adhesion experimental replicate consists of $n = 6$ technical replicates (averaged). Statistics comparing fold change in absorbance at 620 nm (relative to SCR) between SCR and combined SIX1 KD replicates for each group were performed using unpaired two-tailed Welch's T-test. **g** Representative images of transwell invasion inserts taken 72 h after plating for A673 SCR, shSIX1 KD1, and shSIX1 KD2 cells treated with either Rb IgG or indicated integrin neutralizing antibodies. Representative images are shown from one of three independent experiments. **h** Quantification of $n = 3$ independent experimental replicates for assay shown in (**g**). Mean ± SEM shown. Statistics were performed using one-way ANOVA with post-hoc Fisher's least significant difference test. **i** Representative images of transwell invasion inserts from 96 h invasion assay for EWS-502 SCR, shSIX1 KD1, and shSIX1 KD2 cells treated with either Rb IgG or indicated integrin neutralizing antibodies. **j** Quantification of $n = 3$ independent experimental replicates for assay shown in (**i**). Mean ± SEM shown. Statistics were performed using one-way ANOVA with post-hoc Fisher's least significant difference test. Each invasion experimental replicate consists of $n = 3$ technical replicates (averaged). Scale bars = 100 μm. Source data are provided as a Source Data file.

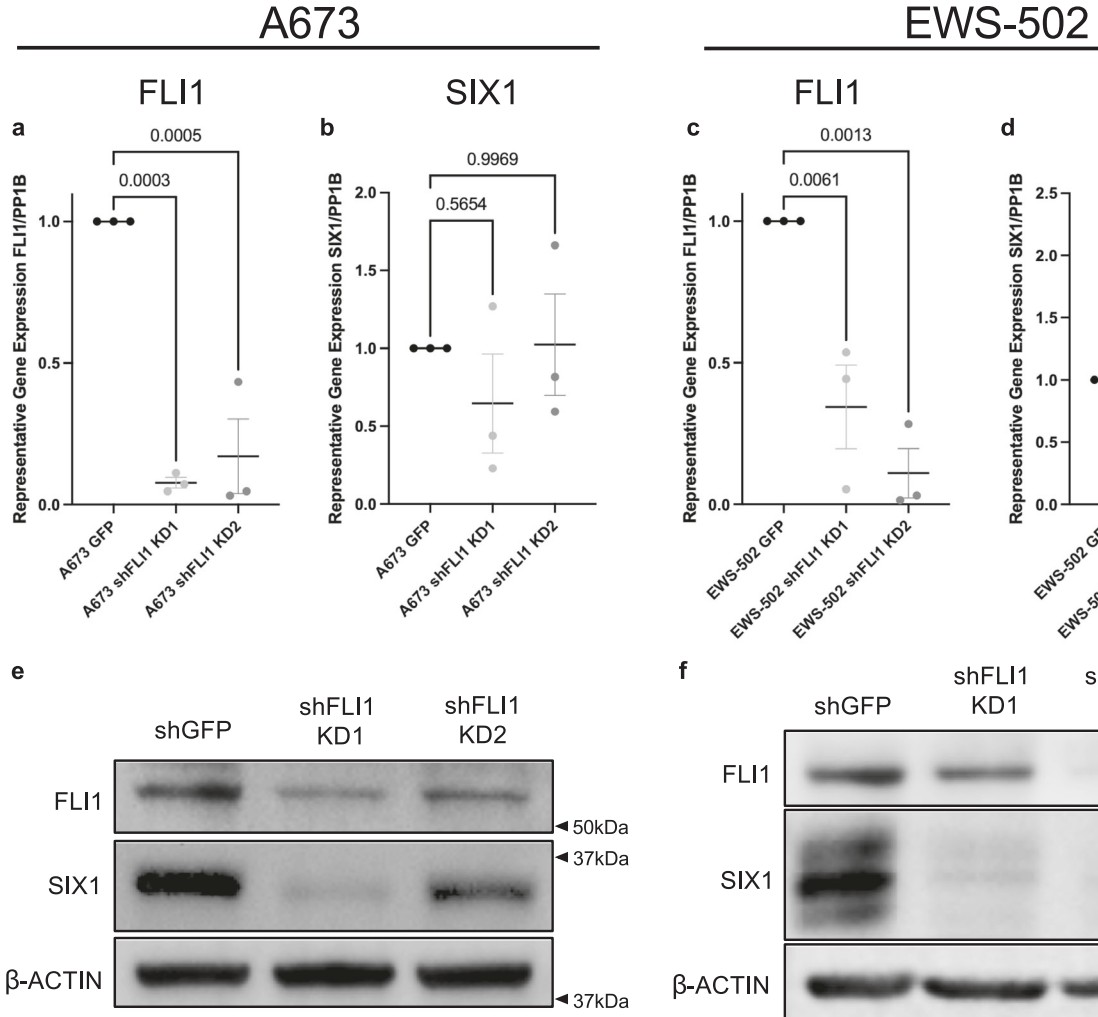

**Fig. 7 | EWS/FLI1 regulates SIX1 protein levels but not transcription.** qRT-PCR expression of **a** EWS/FLI1 and **b** SIX1 in A673 SCR and shFLI1 KD cells. qRT-PCR expression of **c** EWS/FLI1 and **d** SIX1 in EWS-502 SCR and shFLI1 KD cells. For all qRT-PCR experiments, Mean ± SEM of $n = 3$ independent experimental replicates are shown. Each experimental replicate consists of $n = 3$ technical replicates (averaged). Statistics were performed using one-way ANOVA with post-hoc Dunnett's multiple comparisons test. **e** Western blotting analysis of A673 SCR and shFLI1 KD cells for SIX1 levels. **f** Western blotting analysis of EWS-502 SCR and shFLI1 KD cells for SIX1 levels. Western blotting images are representative of three independent experiments. Source data are provided as a Source Data file.

relationship between SIX1 loss and enhanced integrin expression, we asked whether SIX1 may be directly bound with EWS/FLI1 at the promoters of integrins that are most upregulated upon SIX1 KD (Fig. 5). CUT&RUN experiments for SIX1 and FLI1, when compared to H3K4me3 (promoter mark) and H3K4me1 (enhancer mark), previously established in A673 cells[41], showed that both SIX1 and FLI1 occupy the promoters of ITGA1, ITGA2, ITGA4, and ITGA6 (Fig. 8a–d), and also ITGA5 and ITGB1 (Supplementary Fig. 9a, b) in A673 cells. As expected,

SIX1 KD led to reduced SIX1 occupancy at nearly all of these promoter regions relative to SCR samples (Fig. 8a–d).

To confirm the generalizability of these findings, we additionally performed CUT&RUN for SIX1 and FLI1 in the EWS-502 system. In line with what we observed in the A673 model, SIX1 and FLI1 strongly co-localized on integrin promoters in EWS-502 cells (Supplementary Fig. 9c–f). As we observed enriched integrin expression with SIX1 KD in both models, we hypothesized SIX1 is causing gene repression at these

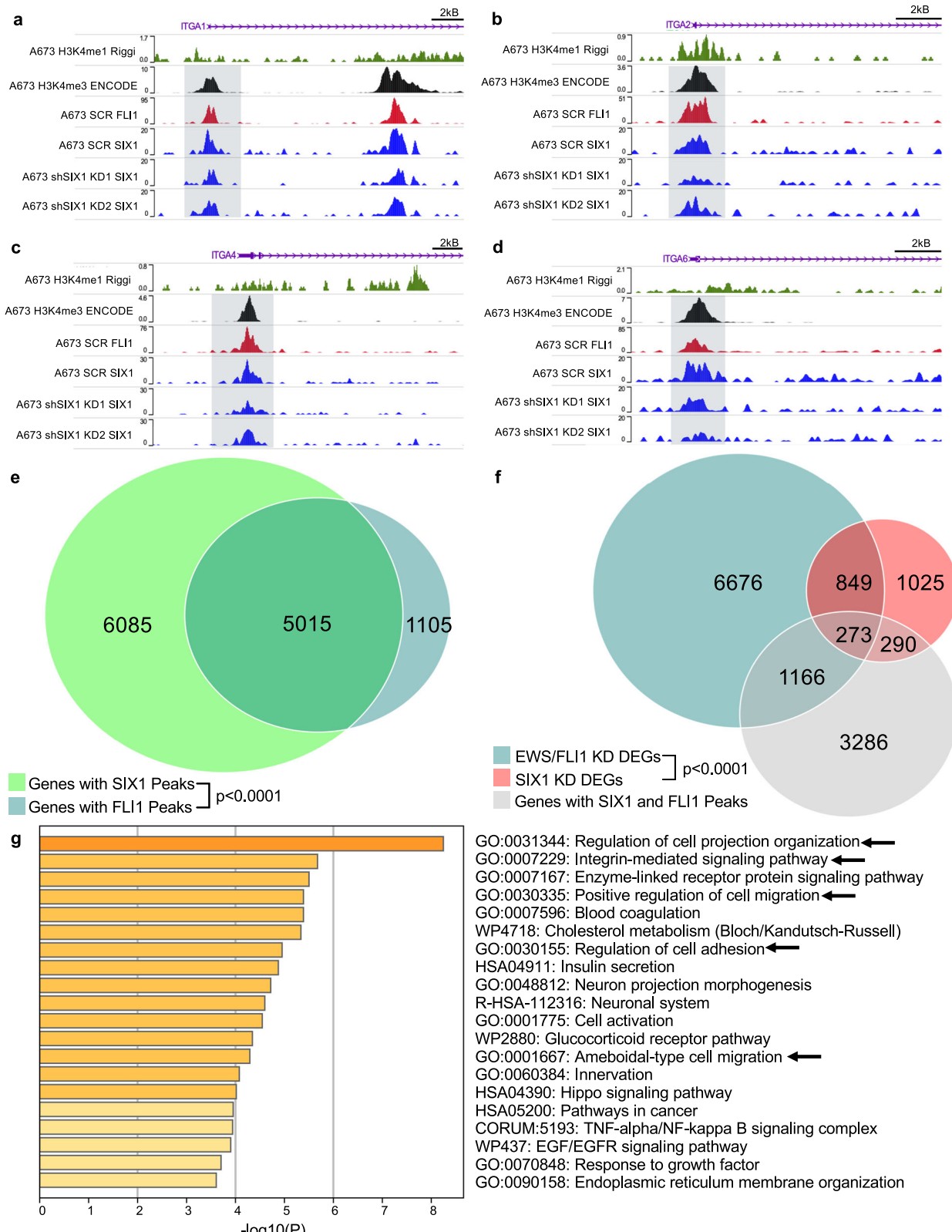

**Fig. 8 | SIX1 and EWS-FLI1 co-bind and co-regulate integrin genes and display genome-wide co-regulation of metastasis-related gene sets.** CUT&RUN tracks for A673 SCR and SIX1 KD cells around the promoter region of **a** ITGA1, **b** ITGA2, **c** ITGA4, and **d** ITGA6. **e** Venn diagram displaying genes with SIX1 and/or FLI1 peaks. Statistics for Venn diagram overlap were performed using a hypergeometric test (one-tailed Fisher's exact test). **f** Venn diagram displaying overlap of differentially expressed genes in A673 EWS/FLI1 KD cells, A673 SIX1 KD2 cells, and genes with both SIX1 and FLI1 peaks from A673 SCR CUT&RUN. Statistics for Venn diagram overlap were performed using a hypergeometric test (one-tailed Fisher's exact test). **g** Metascape enriched gene sets from the three-way overlapping genes from (**f**). Statistical analysis was performed using a hypergeometric test. NGS data is available at GEO GSE215416.

loci. Thus, we additionally performed CUT&RUN for the repressive mark H3K27me3 to look for evidence of de-repression with SIX1 loss. Interestingly, we observed a strong reduction in H3K27me3 enrichment at these loci (Supplementary Fig. 9c–f), demonstrating that these integrins exhibit epigenetic evidence of de-repression with SIX1 KD.

### SIX1 and EWS/FLI1 globally occupy the same gene promoters and co-regulate pro-metastatic gene sets

Due to the observed overlapping binding of SIX1 and EWS/FLI1 on all queried integrins, we asked whether SIX1 and EWS/FLI1 are binding to similar loci genome-wide. In A673 CUT&RUN experiments, SIX1 and EWS/FLI1 were co-bound on 5,015 genes ($p < 0.0001$), which accounts for 81.9% of the genes bound by EWS/FLI1 and 45.2% of the genes bound by SIX1 (Fig. 8e). We observed similarly strong overlap in the EWS-502 system (Supplementary Fig. 10a). This profound overlap in bound genes suggests that SIX1 regulates similar sets of genes as EWS/FLI1 in ES. Indeed, when cross-referencing EWS/FLI1 KD DEGs in A673 cells[41] with the DEGs identified in our A673 SIX1 KD2 RNAseq (Fig. 8f), we observed that 1,122, or 46.0% ($p < 0.0001$), of DEGs upon SIX1 KD also were differentially expressed with EWS/FLI1 KD (Fig. 8f), demonstrating considerable overlap in the transcriptional targets of SIX1 and EWS/FLI1. Of those, 273 genes (24.3%) were bound by both SIX1 and FLI1 in our CUT&RUN dataset (Fig. 8f). Thus, a large number of these genes are likely direct regulatory targets of both SIX1 and EWS/FLI1. We observed a similar overlap between co-bound genes and SIX1-regulated genes in the EWS-502 system (Supplementary Fig. 10b), and also a significant overlap between SIX1 KD DEGs (combined KD) in the EWS-502 system with EWS/FLI1 KD DEGs from the A673 system as described above (Supplementary Fig. 10c), demonstrating that a co-regulated gene network is generalizable across multiple systems of ES.

In addition, gene set enrichment analysis using Metascape[42] on genes that are bound by both SIX1 and EWS/FLI1, and are co-regulated by the two proteins in the A673 system (Fig. 8f), uncovered numerous metastasis-associated gene sets including "positive regulation of cell migration" and "integrin-mediated signaling pathways" (Fig. 8g). Through this same analysis, we observed enrichment for regulation of adhesion-related gene sets in the EWS-502 system, suggesting similar functions in both cell lines (Supplementary Fig. 10d). Additionally, co-binding of SIX1 and EWS-FLI primarily occurs in promoters and regions close to the TSS in both systems (Supplementary Fig. 11a–d), suggesting that genes bound by both SIX1 and EWS/FLI1 are likely to be direct regulatory targets of SIX1 and/or EWS/FLI1, rather than non-specific binding.

Given the dramatic genome-wide co-localization of SIX1 and EWS/FLI1, we asked whether SIX1 and EWS/FLI1 may directly interact, suggesting that one of the proteins may recruit the other to co-bound regions. However, via co-immunoprecipitation we did not observe any direct binding between the two proteins (Supplementary Fig. 12), suggesting that both proteins are binding at shared loci independently.

### SIX1 and EWS/FLI1 both bind to ETS-TF motifs in ES cells

As SIX1 and EWS/FLI1 do not directly interact, we asked whether the profound co-localization of SIX1 and EWS/FLI1 in ES cells was due to the targeting of similar genetic motifs. To this end, we performed motif enrichment analysis on the SIX1 and FLI1 CUT&RUN tracks for A673 SCR and EWS-502 SCR cells to determine which motifs are enriched in the binding sites of both proteins. In both the A673 and EWS-502 systems, FLI1 and SIX1 bound sites were both predominantly enriched with ETS-TF family motifs (Supplementary Tables 1–4). These results demonstrate that SIX1 is preferentially binding at ETS-TF target sites in ES cells, rather than at canonical SIX binding sites, promoting genome-wide co-localization of SIX1 with EWS/FLI1.

### SIX1 and EWS/FLI1 coordinately regulate integrin gene expression in ES

Given the profound overlap in co-bound and co-regulated genes by SIX1 and EWS/FLI1, we asked whether SIX1 and EWS/FLI1 may cooperatively regulate subsets of pro-metastatic genes. To address this question, we used an siRNA targeting the fusion sequence of EWS/FLI1[43] to transiently reduce EWS/FLI1 expression in A673 SCR and shSIX1 KD cells. We then performed qRT-PCR looking at the expression of a panel of integrins genes that we demonstrated are regulated by SIX1 (Fig. 5a, b). After confirming KD of both SIX1 and EWS/FLI1 (Supplementary Fig. 13a, b), we queried for changes in integrin expression. For ITGA1, ITGA6, and ITGB1, independent KD of SIX1 or EWS/FLI1 led to similar levels of enhanced gene expression, and combined KD of SIX1 and EWS/FLI1 led to similar or modest additional increased expression over each independent KD alone (Supplementary Fig. 13c–e). These data suggest that both EWS/FLI1 and SIX1 are required for the repression of these integrins and that they likely cooperate at these loci. In contrast, ITGA4 and ITGA5 showed a modest increase in expression of with SIX1 KD, however, the expression of these integrins was unchanged or even slightly reduced with EWS/FLI1 KD, suggesting that EWS/FLI1 expression may actually maintain the expression of these two integrins in some contexts (Supplementary Fig. 13f, g). These divergent patterns of regulation suggest a complex interplay between SIX1 and EWS/FLI1 in the regulation of integrin genes, which suggests nuanced, gene-specific, co-regulatory mechanisms for EWS/FLI1 and SIX1 in influencing gene expression and metastasis.

### Higher SIX1 expression is associated with improved event-free survival in ES patients

Our in vitro and in vivo findings suggest that SIX1 may repress tumor progression. To investigate the clinical relevance of these findings, event-free survival (EFS) was examined in a cohort of 85 ES patients[44]. In line with our experimental findings, high SIX1 expression is significantly correlated with improved EFS in this cohort (Fig. 9a). There are very few publicly-available ES human datasets containing both gene expression and outcome data, and these datasets are often constrained by low sample size, limiting the scope of this analysis to a single dataset. Nonetheless, these data suggest that SIX1 expression may be prognostic in ES. Taken together, our data suggest the model (Fig. 9b) that the presence of SIX1 on promoters of integrins and other pro-metastatic genes typically defined by the presence of ETS motifs, likely in conjunction with EWS/FLI1, leads to repressed integrin/pro-metastatic gene expression, and promotes reduced invasiveness and decreased metastasis for high-SIX1-expressing ES cells. Decreased levels of SIX1, secondary to decreased EWS/FLI, would thus lead to de-repression of these genes, leading to pro-metastatic gene expression and increased metastatic propensity of ES cells.

## Discussion

In numerous cancer contexts, SIX1 positively regulates proliferation, tumor-initiating-cell characteristics, and migration and invasion, resulting in tumor progression and metastasis[10–12,17,18,22,45]. However, in the context of ES, we demonstrate a metastasis-suppressive function for SIX1. We hypothesize that this function of SIX1 may be specific to the genetic background of ES, due to the considerable overlap in direct transcriptional regulatory targets between SIX1 and the oncogenic fusion protein EWS/FLI1[46], and the predilection of SIX1 for binding at ETS transcription factor binding motifs in ES cells. The work outlined in this study underscores the importance of understanding tumor context when examining gene function.

Metastasis in cancers of epithelial origin is frequently associated with tumor cells undergoing an epithelial-to-mesenchymal transition (EMT), losing cell-to-cell attachment, and enhancing motility, invasiveness, and resistance to apoptosis[32,47]. SIX1 expression in

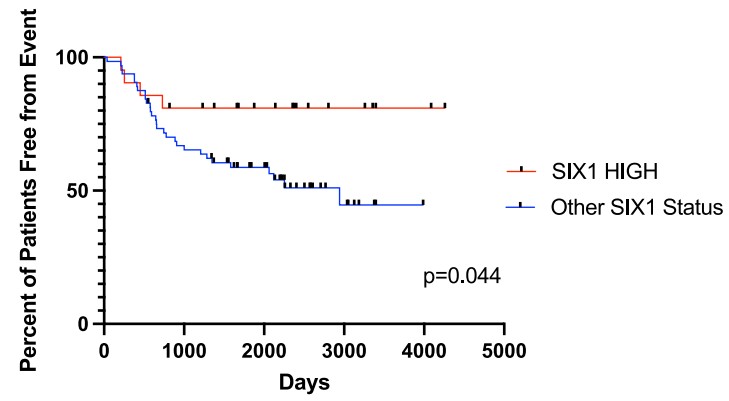

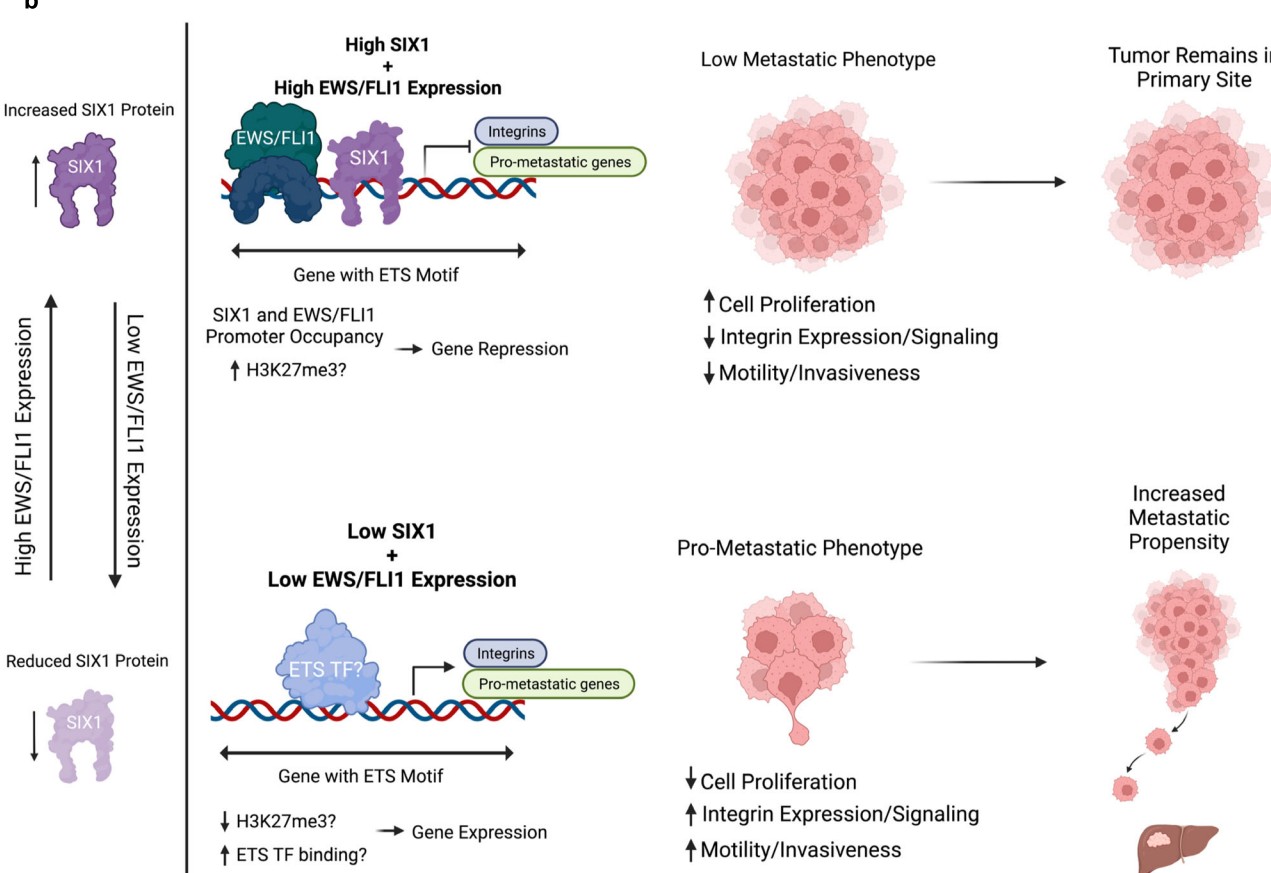

**Fig. 9 | High SIX1 expression correlates with improved Event-Free Survival.**
**a** Event-free survival of n = 85 patients with ES tumors expressing high SIX1 (top 25%) or other SIX1 expression status (bottom 75%). Statistics were performed using log-rank test. **b** Model of SIX1 expression in ES tumors and the associated pro- or anti-metastatic phenotypes. Model created with Biorender.com. Source data are provided as a Source Data file.

carcinomas has consistently been shown to promote EMT and enable epithelial-derived carcinoma cells to adopt a mesenchymal-like phenotype, characterized by reduced expression, or relocalization, of epithelial markers, such as E-cadherin, and increased expression of mesenchymal markers, such as N-cadherin[32]. The pro-metastatic functions of SIX1 in carcinomas have been primarily linked to this EMT-promotional role[10,11,18]. In sarcomas, which are derived from the mesenchyme, tumor cells already possess cellular phenotypes necessary for metastasis, and therefore the role of an EMT or EMT-like process in this context is less clear. ES represents an interesting case study for trying to understand this concept due to many of these EMT-

associated phenotypes being under the direct control of EWS/FLI1[5]. ES tumors often display a high level of cell-to-cell heterogeneity, varying widely in the expression level of EWS/FLI1 within a single tumor[5]. EWS/FLI1-low cells have a higher metastatic propensity primarily due to cell-intrinsic factors such as enhanced motility from increased actin cytoskeleton reorganization[5,6], reduced expression of cell-cell adhesion proteins[5,6], and enhanced expression of cell-matrix adhesion proteins[5–7,48], many of which mirror phenotypes associated with epithelial cells that have undergone EMT.

In the present study, we found that SIX1 expression promotes growth in soft agar while simultaneously repressing migration,

invasion, and metastasis in ES cells, diverging from phenotypes typically associated with higher SIX1 expression in other cancer types[11–13,15,18,22]. Our data indicate that while SIX1 KD is associated with a reduction in EMT gene signatures in ES, SIX1 KD ES cells have reductions in both E-cadherin and N-cadherin expression, and SIX1 drives mixed effects on apoptosis- and chemo-resistance, a divergence from the bona fide EMT driven by SIX1 in carcinomas. Our data suggest that SIX1 KD cells may actually be more plastic/less differentiated as they do not clearly polarize as epithelial or mesenchymal cells, and as such, the pro-metastatic phenotype of these cells cannot be sufficiently explained through SIX1-regulated shift towards a more mesenchymal-like state.

Because SIX1 does not regulate a canonical EMT in this context, we reasoned that the role of SIX1 is unique to ES due to the expression of the EWS/FLI1 fusion protein. In line with this idea, we observed striking overlap in the phenotypes associated with EWS/FLI1-low expressing cells and SIX1 KD cells, including reduced growth rates, enhanced motility, enhanced invasiveness, increased integrin expression and signaling, and a high propensity for metastasis[5–7]. Consistent with these shared phenotypes, SIX1 and EWS/FLI1 displayed profound overlap in genome-wide binding localization and direct transcriptional regulatory targets, many of which enrich for metastasis-associated gene sets, demonstrating that the two proteins are co-regulating metastatic gene expression in this context. This finding begs the question as to why, in the context of ES, SIX1 adopts this particular regulatory pattern that is distinct from its role in other tumor types. It has been previously shown that EWS/FLI1 expression in ES cells causes genome-wide chromatin remodeling[41,49], specifically around sites containing GGAA or canonical ETS motifs[41], which are associated with functional changes in gene expression. As opposed to binding at its canonical DNA binding motif, SIX1 may be binding at unique loci containing ETS motifs in the context of ES due to increased accessibility of chromatin at these sites with EWS/FLI1 expression, or possibly decreased accessibility of canonical SIX1 binding sites. Of note, while never before observed in cancer, it was recently shown that SIX1 binds at loci enriched for ETS motifs in the context of auditory sensory epithelial cells[50], demonstrating that SIX1 has the capacity to bind at these loci in specific contexts. Interestingly, a mechanism previously proposed for the repressive functions of EWS/FLI1 in ES is the displacement of endogenous wild-type ETS TFs from ETS binding sites by EWS/FLI1[41], demonstrating that at specific loci, simply the localization of EWS/FLI1 is sufficient to prevent gene activation without active repression, and it is possible that this steric repression is amplified by SIX1 co-binding. However, because we observe enrichment of H3K27me3 at the promoters of integrins in the EWS-502 SCR cells relative to SIX1 KD cells, our data suggest epigenetic means of repression may also be playing a role in this context. As ETS TFs have been implicated in promoting tumor progression[51], in part through transcriptional regulation of integrins and ECM remodeling components[52], SIX1-mediated repression of ETS TF target genes, in concert with EWS/FLI1, could explain SIX1's anti-metastatic role in this context.

Given the sufficiency of SIX1 KD to promote EWS/FLI1-low-like phenotypes in this context, and the observation that EWS/FLI1 stabilizes SIX1 protein levels, it begs the question as of whether SIX1 is a critical downstream effector of many EWS/FLI1-driven phenotypes in ES. Our data looking at independent or combined KD of SIX1 and EWS/FLI1 on integrin genes suggest a complex interplay in the regulatory functions of SIX1 and EWS/FLI1 on metastasis-associated genes, displaying elements of redundant and cooperative repression, and possibly even antagonism at some loci. These findings suggest that the exact expression of pro-metastatic genes in ES is likely dependent on the expression level of both SIX1 and EWS/FLI1, and the resulting cellular phenotype can be precisely tuned by the relative level of each protein. Nonetheless, our findings suggest that SIX1 KD alone can

repress ES metastasis, and therefore the pro-metastatic phenotype associated with EWS/FLI1-low cells is likely amplified by the concomitant reduction in SIX1 protein levels.

In sum, these findings demonstrate that SIX1 and EWS/FLI1 co-regulate an anti-metastatic gene network in ES, and reduced expression of SIX1 is sufficient to promote migration, invasion, and metastasis, likely in part through integrin upregulation and increased integrin signaling. Additional studies will be necessary to determine the exact mechanisms behind this genomic and regulatory pattern of SIX1 in ES, and any additional characteristics of the ES context beyond EWS/FLI1 expression which may be responsible for this anti-metastatic function of SIX1. While SIX1 complex inhibitors (targeting either SIX1 or its critical cofactor, EYA proteins) are currently in development for the treatment of other tumor types[10,24,25], these findings suggest that targeting of SIX1 in ES may actually promote metastasis and tumor progression, and emphasize the importance of additional studies into the role of SIX1 in different sarcomas or fusion-driven tumors to determine the generalizability of these findings to similar tumor types.

## Methods
The work compiled here complies with Institutional Biosafety Committee (Protocol #1140) regulations. All animal studies were performed according to protocols reviewed and approved by the Institutional Animal Care and Use Committee (IACUC) at the University of Colorado AMC (Protocol #00089).

### Cell lines and culture conditions
Ewing Sarcoma cell lines A673, EWS-502, SK-ES-1, SK-N-MC, and hMSC (Lonza) cells were cultured as previously described[53]. TC71 cells were grown in RPMI base media. All EWS cell line media was supplemented with 10% fetal bovine serum (FBS), 1% Pen/Strep, 10 mM Hepes, 1% non-essential amino acids (NEAAs), and 1 mM Sodium Pyruvate. A673 A1C media was additionally supplemented with 200 μg/ml zeocin and 20 μg/ml blasticidin. All cell lines were authenticated in 2015[48], and A673 and EWS-502 SCR and SIX1 KD cells were re-authenticated via STR profiling in 2023. All cells were checked for mycoplasma every 3-6 months to ensure the accuracy of our findings. Cell lines were passaged at least 2 times after thawing prior to use in experiments, and were re-thawed periodically to maintain low passage numbers. Of note, SK-N-MC is commonly misidentified as a human neuroblastoma cell line as per the ICLAC Register of Misidentified Cell Lines, but it is being used as a model for its correct cell type, Ewing's sarcoma, in this manuscript.

### Generation of knockdown cell lines
Lentiviral shRNA constructs were used for stable knockdown of SIX1 in EWS cell lines according to manufacturers' protocols. PLKO.1-derived SIX1 shRNA constructs TRCN0000015234, TRCN0000015235, and TRCN0000015236 were obtained from The University of Colorado Cancer Center Functional Genomics Core Facility through a contract with Sigma. Stable A673 shSIX1 KD lines were generated with TRCN0000015234 (KD1) and TRCN0000015236 (KD2). Stable EWS-502 shSIX1 KD lines were generated with TRCN0000015234 (KD1) and TRCN0000015235 (KD2). Stable shFLI1 KD lines were generated with shGFP or TRCN0000005322 (KD1) and TRCN0000005324 (KD2) constructs from Open Biosystems as previously described[53]. Lentivirus was generated through transfection of HEK293T cells with the aforementioned constructs along with pMD2.G and psPAX2 lentiviral packaging plasmids using FuGene HD Transfection Reagent (Promega, E2312) or Lipofectamine 2000 Transfection Reagent (Invitrogen, 11668019). Viral particles were collected at 48 h post-transfection, sterile filtered with a 0.45μm filter syringe, and treated with 10 μg of polybrene prior to transducing target cells. 24 hours post-viral infection, A673 and EWS-502 shSIX1 and shFLI1 KD cells were selected with 5.0 μg/ml and 1.0 μg/ml of puromycin, respectively, and were

maintained in these puromycin concentrations for maintenance. The doxycycline-inducible EWS/FLI1 shRNA A673-shA1C cells were generated in the lab of Dr. Olivier Delattre and cultured as previously described[54]. 100 nM of ON-TARGET plus SMARTpool siRNA constructs [siRNA control (D-00110-05-20) and siSIX1 (L-020093-00-0005; Dharmacon)] were used for transient KD of SIX1 in the EWS-502 system.

## RNA expression analysis

Total RNA was extracted from cells at 80% confluency using the RNeasy Plus Mini RNA Isolation Kit (Qiagen, 74136) according to the manufacturer's instructions. cDNA synthesis was performed using iScript cDNA Synthesis Kit (Biorad, 1708891) from 1 μg of mRNA. qRT–PCR assays were performed using ssoFast Evagreen Supermix (BioRad, 1725205) and run and analyzed using the Biorad CFX96 qPCR instrument. See Supplementary Table 5 in the Supplementary Data file for all primer sequences used.

## Immunoblot analysis

Whole-cell protein lysates (WCL) were generated as previously described[12]. Membrane proteins were extracted using the Mem-PER™ Plus Membrane Protein Extraction Kit (Thermo Scientific 89842) according to the manufacturer's instructions. For WCL and membrane immunoblots, 25-50 μg of protein were electrophoresed on 5% stacking and 10% running gel and transferred to PVDF membranes. The membranes were blocked in 5% milk or bovine serum albumin (BSA) in TBST for 1 hr and incubated with primary antibody, diluted in 5% BSA in TBST, at 4 °C overnight. The blots were then washed with TBST and incubated for 1 hour at RT in secondary antibody (1:10,000 dilution in 5% BSA in TBST) prior to imaging. The blots were developed using SuperSignal West Pico Chemiluminescent Substrate (Thermo Scientific, 34080) and/or SuperSignal West Femto Chemiluminscent Substrate (Thermo Scientific, 34096) depending on the strength of the signal. Blots were imaged on the Licor Odyssey FC using Licor Image Studio (v. 5.2.5) software and analysis was performed with ImageJ (v1.53). For re-imaging of different proteins at similar molecular weights, blots were stripped with Restore Western Blot Stripping Buffer (Thermo Scientific, 21059) for 5 minutes at RT, then were blocked and re-probed as described above. All uncropped blots in the main figures can be found in the Source Data file. All uncropped blots for supplementary figures can be found at the end of the Supplementary Data file. See Supplementary Table 6 in the Supplementary Data file for all primary antibody information. See Supplementary Table 7 in the Supplementary Data file for all secondary antibody information.

## IncuCyte cell growth assays

Cell growth was measured using IncuCyte Zoom (Essen Biosciences) Live-Cell Analysis Platform. 5000 A673 or EWS-502 cells were plated in 5 replicates in a 96-well plate and were imaged every 2 hours with a 4x objective. The percent confluence of each well at each time point was calculated using IncuCyte Zoom image processing software. All 5 replicates per group were averaged to provide a single average confluence per time point, and each group was normalized to a consistent confluence percentage (as defined as being within 1% point) set to time point 0 h for all groups. Cell growth curves were generated by graphing average confluence and standard deviation across technical replicates for each group for each time point. Statistical differences between groups were calculated using a longitudinal mixed effects model that compares groups over repeated measures.

## Anchorage-independent growth assays

6-well plates were coated with 2 ml of bottom agar solution (0.5% Noble agar [BD Difco 214230], 20% FBS, 1X DMEM, 2 μg/ml puromycin) and allowed to solidify at room temperature (RT). 10,000 cells per

group in triplicate for A673 SCR, A673 shSIX1 KD1, and A673 shSIX1 KD2/EWS-502 SCR, EWS-502 shSIX1 KD1, and EWS-502 shSIX1 KD2 were added to 3 ml top agar solution (0.38% Noble agar, 20% FBS, 1X DMEM, 2 μg/ml puromycin), per well. Three ml of top agar solution containing 10,000 cells was plated on top of the bottom agar solution in each well. The top agar solution was allowed to solidify at room temperature (RT) before adding 2 ml of complete media per well. A673 and EWS-502 cells were grown in anchorage-independent conditions in an incubator at 37 °C (5% $CO_2$) for 9 days and 11 days, respectively, replacing the media every 2 days, before staining. Cells were stained with 200 μl of 1 mg/ml Nitroblue Tetrazolium Chloride (Amresco, 0329-1 G) for 48 h in the incubator. After staining, plates were imaged on the GE ImageQuant LAS 4000, and images were analyzed and quantified using ImageJ.

## Cell migration and invasion assays

For transwell cell migration assays, A673 SCR, shSIX1 KD1, and shSIX1 KD2 cells and EWS-502 SCR, shSIX1 KD1, and shSIX1 KD2 cells were resuspended in 200ul/insert of serum-free media and transferred into cell culture inserts with 8 μm pores (BD Falcon, 353097) (250,000 cells/triplicate inserts per group). Cell culture inserts were placed in 24-well companion dishes (Corning, 353504) containing 800 μl of full serum media per well and put in the incubator at 37 °C (5% $CO_2$) for 24 h (A673) or 40 hours (EWS-502). After incubation, media inside of the insert was aspirated, and the bottom of each insert was fixed in 4% PFA for 10 minutes at RT. After fixation, the bottom of each insert was stained with 0.1% crystal violet solution for 45 minutes at RT and then washed with ddH2O to remove excess stain. Inserts were dried at RT for 24 h prior to imaging on an Olympus CKX41 microscope. Three representative images of each insert were taken at 10X magnification (imager was blinded to the treatment group of each insert). Images were analyzed and quantified using ImageJ. This protocol is also followed for invasion assays with the exception of the addition of 75 μl Matrigel (BD Biosciences, 354480) (1:20 diluted in serum-free media) to the top of each insert prior to adding cells. For invasion assays with the A673 cell line, 200,000 cells/insert were used with an incubation time of 72 h prior to staining. For invasion assays with the EWS-502 cell line, 150,000 cells/insert were used with an incubation time of 96 h prior to staining. Scratch wound cell migration assays were performed as previously described[29].

## In vivo experimental metastasis assays and subcutaneous tumor growth assay

All animal studies were performed according to protocols reviewed and approved by the Institutional Animal Care and Use Committee (IACUC) at the University of Colorado AMC. A673 SCR, shSIX1 KD1, and shSIX1 KD2 cells were tagged with firefly luciferase and were imaged prior to in vivo injection to determine consistent tagging. One million cells per mouse were re-suspended in serum- and antibiotic-free media and injected in the tail vein of three- to twelve-week-old male NOD/SCIDγ (NSG) mice. All mice used in all experiments were bred and maintained in-house at the CU AMC Vivarium by staff at the CU Office of Laboratory Animal Resources (OLAR). N = 6 mice were used per group. Mice were imaged weekly with the IVIS200 bioluminescent imaging system. Prior to imaging, each mouse received an intraperitoneal injection of 100 μl of 100X luciferin (Gold Biotechnology, LUCK-1G). Each mouse was then anesthetized via isoflurane inhalation and imaged at 10 minutes post injection. All mice were sacrificed when they began to show clinical signs of health deterioration as confirmed by an overseeing veterinarian. After sacrifice, tumors and livers were extracted and fixed in 10% neutral-buffered formalin for 24 h, followed by storage in 70% EtOH at 4 °C. Imaging data was processed and normalized using LivingImage (v. 4.5.2) software. For the A673 system, statistical differences between groups for luciferase intensity were calculated using a two-way ANOVA with Dunnett's post-hoc multiple

comparisons test. Survival curves were compared using log-rank (Mantel-Cox) analysis. For the EWS-502 system, 400,000 cells per mouse were resuspended in serum and antibiotic-free media and injected in the tail vein of three- to twelve-week-old male NSG mice. Samples sizes were as follows (EWS-502 SCR N = 5 mice, EWS-502 shSIX1 KD1 N = 5 mice, EWS-502 shSIX1 KD2 N = 4 mice [one mouse died during tail vein injection]). After 33 days, the mice were sacrificed and necropsy was performed on each. H&E was performed on representative liver sections as previously described[55]. Standard light microscope images were taken at indicated magnification. Histologic examination of liver sections was performed by a board-certified pathologist (PJ). For the subcutaneous tumor growth model, one million cells per mouse were re-suspended in serum and antibiotic-free media (no Matrigel) and injected into the flank of three- to twelve-week-old male and female NSG mice. Sample sizes used were as follows: A673 SCR N = 6, A673 shSIX1 KD1 N = 5, A673 shSIX1 KD2 N = 4 – one mouse was excluded from analysis after being determined to be a significant outlier via Grubbs' test. Tumors were measured with calipers at indicated intervals and volume was calculated as [long axis (cm) *short axis$^2$ (cm)]/2. Statistical differences between groups are calculated using longitudinal mixed-models analysis which compares differences between groups over time. Housing conditions for animals used in this study: Light cycle: 14 h light: 10 h dark; Temperature: 72 °F +/− 2 °F; Humidity: 40% +/− 10%; Water: Hyperchlorinated (2-5 ppm) Reverse Osmosis delivered via automatic watering; Food: Teklad (Envigo) diets; Rodents - Standard diet (2920X). Breeder diet (2919) both irradiated. Of note, the maximal tumor size permitted by the Colorado AMC IACUC is 2 cm$^2$ which was not exceeded in any study in this manuscript.

## RNA-Sequencing Analysis
A673 SCR, shSIX1 KD1, and shSIX1 KD2 and EWS-502 SCR, shSIX1 KD1, and shSIX1 KD2 RNA samples were extracted using the RNeasy RNA Isolation Kit (Qiagen) in triplicate and were sent to the University of Colorado Cancer Center Genomics Core Facility for sample preparation and sequencing and generation of raw data. Library preparation was performed using the NuGEN Universal Plus mRNA-Seq kit. Samples were sequenced with paired-end reads (40 million reads/sample) on the Illumina NovaSEQ6000 sequencing platform. The quality of the fastq files was assessed using FastQC[56] and MultiQC[57]. Illumina adapters and low-quality reads were filtered out using BBDuk (http://jgi.doe.gov/data-and-tools/bb-tools). Trimmed fastqc files were aligned to the hg38 human reference genome and aligned counts per gene were quantified using STAR[58]. Transcript expression levels were estimated with Salmon using inferential replicates[59]. Differential gene and transcript analyses were performed using the DESeq2[60] and Swish[61] packages. After processing, the University of Colorado Biostatistics and Bioinformatics Shared Resource RNA-seq analysis tool was used to perform pathway analysis, including over-representation analysis, functional gene set enrichment analysis (GSEA) and Kegg Topology analysis. A fold change (FC) cutoff of 2x was used for all differential gene expression analysis.

## Propidium Iodide staining and flow cytometry
Cells were harvested through trypsinization, spun down, and washed 3x with PBS supplemented with 2% FBS. Cells were incubated with 50 µg/mL propidium iodide (PI) (Sigma-Aldrich, 25535-16-4) in PBS plus 2% FBS at 4 °C in the dark for 30 minutes. After incubation with PI, cells were washed 3x with PBS plus 2% FBS, passed through a cell strainer (Falcon, 352235), then analyzed by flow cytometry on a BD Accuri instrument. Flow cytometry data was analyzed using FlowJo.

## Drug sensitivity assays
A673 and EWS-502 SCR and SIX1 KD cells (5000 cells per well) were plated in 5 replicate wells per treatment dose in a white bottom 96-well plate. After 24 h, cells were treated with increasing doses of Vincristine (Sigma-Aldrich, V8388) or Actinomycin D (Sigma-Aldrich, A1410) and incubated at 37 °C for 48 h. After incubation, 100ul of CellTiter-Glo 2.0 (Promega, G9243) was added per well and the plate was rocked in the dark for 10 minutes. Luminescence was measured using the SpectraMax iD3 Microplate Reader (Molecular Devices). Doses used were vehicle (0 nM), 0.1, 0.5, 1.0, and 5.0 nM for A673 Vincristine and vehicle (0 nM), 1.0, 2.5, 5.0, and 10.0 nM for EWS-502 Vincristine and Actinomycin D for both cell lines. IC50 values were calculated using nonlinear regression (four parameters) on normalized values using Prism software (v9.0; GraphPad). Replicate IC50 values per group were compared using a One-way Analysis of Variance (ANOVA) with Dunnett's post-hoc test for multiple comparisons.

## Integrin-blocking invasion assays
Integrin neutralizing invasion assays were performed as described above with the following changes. 200,000 cells were plated per insert for both A673 and EWS-502 cell lines. Neutralizing antibodies or IgG control antibodies were added to each cell line prior to plating. See Supplementary Table 7 in the Supplementary Data file for all neutralizing antibody information.

## CUT&RUN
CUT&RUN extraction and library preparation was performed as previously described[12] using the antibodies described in Supplementary Table 8 in the Supplementary Data file.

## CUT&RUN and overlapping CUT&RUN/RNA-seq data processing and analysis
The quality of the fastq files was accessed using FastQC[56] and MultiQC[57]. Illumina adapters and low-quality reads were filtered out using BBDuk (http://jgi.doe.gov/data-and-tools/bb-tools). Bowtie2 (v.2.3.4.3)[62] was used to align the sequencing reads to the hg38 reference human genome. Samtools (v.1.11)[63] was used to select the mapped reads (samtools view -b - q 30) and sort the bam files. PCR duplicates were removed using Picard MarkDuplicates tool (http://broadinstitute.github.io/picard/) The normalization ratio for each sample was calculated by dividing the number of uniquely mapped human reads of the sample with the lowest number of reads by the number of uniquely mapped human reads of each sample. These normalization ratios were used to randomly sub-sample reads to obtain the same number of reads for each sample using samtools view -s. Bedtools genomecov was used to create bedgraph files from the bam files[64]. Bigwig files were created using deepTools bamCoverage[65] and visualized using the WashU Epigenome Browser. Peaks were called using MACS2 (v2.1.2)[66]. IDR was used to identify the reproducible peaks between the replicates[67]. Further processing of the peak data was performed in R, using the following tools: valR[68], DiffBind, and ngs.plot[69].

## Motif analysis
All motif analyses were performed using the findMotifsGenome.pl program in the HOMER package (v.4.101.1)[70]. The motifs were ranked by the -log p-value. (Top 20 motifs are shown).

## Immunoprecipitation
Immunoprecipitation experiments were performed for SIX1 as previously described[10] using 2 µg of Rb anti-SIX1 antibody (Cell Signaling Technology, D4A8K). Immunoblotting was performed as described above using Rabbit TrueBlot anti-rabbit IgG HRP secondary antibody (Rockland Immunochemicals, 18-8816-31) when probing for FLI1 and SIX1.

## Cell adhesion assays
96-well plates coated with Collagen IV, Fibronectin, and Laminin (Corning) were blocked with 1% BSA in 1XPBS for 1 hour at RT.

Following blocking, 100,000 cells/well (EWS-502 – all assays) (A673 – Laminin adhesion) or 200,000 cells/well (A673 – Fibronectin/Collagen IV) were added to plates in replicates of 6 per group (SCR, shSIX1 KD1, and shSIX1 KD2). Plates were incubated at 37 °C for 1 h. Plates were washed with PBS and fixed with methanol prior to staining with 0.05% crystal violet in ddH2O for 40 mins. Plates were washed with PBS and dye was solubilized in 10% glacial acetic acid. Absorbance per well was measured at 620 nm on a Biotek Synergy 2 microplate reader.

### Combined SIX1 and EWS/FLI1 knockdown experiments

A673 SCR or SIX1 KD cells were treated with 30 nM control non-targeting siRNA (Dharmacon) or siEWS/FLI1 custom siRNA designed to target the EWS-FLI1 fusion sequence[43] (5' GGC AGC AGA ACC CUU CUU A-dCdG, Horizon Discovery) with RNAi Max transfection reagent (Invitrogen, 13778075) for 12 h at 37 °C. Media was then replaced with fresh A673 media. At 48 h post-transfection, RNA was extracted and cDNA was synthesized as described above. Expression of integrins was assessed via qRT-PCR using SIX1, EWS/FLI1, and integrin primer sets as described above.

### Analysis of human datasets for association of Event-Free Survival and SIX1 expression

Exon array data for 85 Ewing sarcoma human tumor samples were downloaded from the NCBI Gene Expression Omnibus (GEO ID: GSE63157) after generation as described previously[44]. Tumor-derived SIX1 gene expression and event-free survival (in days +/- event) were mapped together for each patient for subsequent analysis. Event-free survival was plotted for SIX1 high expressers (Top 25% of SIX1 expression n = 21 patients) vs. SIX1 low expressers (Bottom 75% of SIX1 expression n = 64 patients). Statistical differences between groups was calculated by log-rank (Mantel-Cox) test.

### Model

Created with BioRender.com.

### Statistics

Prism software (v9.0; GraphPad) was used for all statistical analyses with the exception of longitudinal mixed-model analyses and hypergeometric tests described below. In all in vitro experiments, the conditions were run with at least 3 technical replicates and repeated at least 3 independent times. Two-tailed unpaired Welch's T-test was used to compare SCR and combined SIX1 KD groups. One-way Analysis of Variance (ANOVA) parametric followed by Fisher's Least Significant Difference test was used to compare multiple groups. Alternatively, Bonferroni or Dunnett's post-hoc tests were used to correct for multiple comparisons where applicable (indicated in figure legends). A two-way ANOVA followed by a Dunnett's post-hoc test was used to compare bioluminescence changes in the A673 tail vein experiment. Experiments demonstrating changes over time (cell growth and tumor growth assays) were compared using a longitudinal mixed effects model that compares groups over repeated measures. A hypergeometric test was used to assess the significance of 2-way Venn diagrams. A total gene number of 60,664, including genes for coding and noncoding RNAs, was used for all hypergeometric tests. Specific analyses used for each experiment are described in the figure legends.

### Reporting summary

Further information on research design is available in the Nature Portfolio Reporting Summary linked to this article.

### Data availability

Raw NGS data (RNA-sequencing, CUT&RUN) has been deposited into the Gene Expression Omnibus and are publicly available under the accession number GSE215416 (https://www.ncbi.nlm.nih.gov/geo/query/acc.cgi?acc=GSE215416). The data supporting the findings of this study are available within the paper and its Supplementary Information. Raw data for all experiments are provided in the Source Data file. Publicly available exon array data for 85 Ewing sarcoma human tumor samples (used in Fig. 9a) is available from the NCBI Gene Expression Omnibus (GEO ID: GSE63157 - https://www.ncbi.nlm.nih.gov/geo/query/acc.cgi?acc=gse63157)[44]. Uncropped Western blot images are provided in the Source Data file and in the Supplementary Data file. Source data are provided with this paper.

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

## Acknowledgements

This work has been supported through grants from the NIH: R01 CA224867, CA221282, CA095277 (HLF), R01CA183874 (to PJ), an Alex's Lemonade Stand Foundation Innovation Award and Million Mile Match Funding (HLF), and a training fellowship F30CA257215 (to C.J.H.). This work used the Cell Technologies (RRID:SCR_021982), Genomics (RRID:SCR_021984), Functional Genomics (RRID:SCR_021987), Animal Imaging (RRID:SCR_021980), and Biostatistics and Bioinformatics Shared Resource (RRID:SCR_021983) supported by P30CA046934. We thank the Delattre laboratory for generously providing the A673 shA1C-inducible EWS/FLI1 KD cells. We thank Dexiang Gao for guidance with statistical analyses. We thank Lisa Reeves for her intellectual contributions to this project. We also thank Veronica Wessells and Janet Parrish for providing technical support.

## Author contributions

H.L.F. conceptualized and supervised experiments with input from J.C.C., P.J., and C.J.H. C.J.H., K.M.F., D.N., M.Y.V. and M.J.O. completed cell line experiments and were involved with data generation and interpretation. C.J.H. and V.Z. performed mouse experiments. P.J. scored tumor histology in mouse models. C.J.H. performed RNA-seq experiments. J.Y.H. and A.L.G. performed CUT&RUN experiments. E.P.D. processed NGS data and performed motif analysis. E.P.D. and C.J.H. analyzed RNA-seq and CUT&RUN datasets. E.P.D. and V.S. performed overlapping RNA-seq and CUT&RUN analysis. C.J.H. performed human data analysis. C.J.H. wrote the manuscript with significant input from H.L.F. All authors contributed to manuscript editing and review.

## Competing interests

The authors declare no competing interests.
