## [Peer Review File · Nature Communications]

SIX1 and EWS/FLI1 co-regulate an anti-metastatic gene network in Ewing SarcomaReviewers' Comments:

Reviewer #1:

Remarks to the Author:

Here the authors explore the role of the transcription factor SIX1 in Ewing sarcoma. The authors propose that SIX1 expression, in concert with EWS::FLI1 expression, exerts an anti-metastatic effect in Ewing sarcoma. This is an innovative discovery given that it is contrary to the role of SIX1 in most other cancers. SIX1-knock down reduced anchorage independent growth in vitro, but increased migration in 2D assays and outgrowth after tail vein injection. The authors identified a striking overlap in genes bound by both EWS::FLI1 and SIX1, and propose that a subset of these are co-regulated by both TFs to reduce metastatic potential, partly through repressing a subset of integrins. Integrin blocking antibodies reduced migration of SIX1 knock down cells in 2D revealing that repression of integrins is at least partially responsible for the "anti-metastatic" properties of SIX1 in Ewing sarcoma cells. While these findings are intriguing and partially advance our understanding of pro- and anti-metastatic gene networks in Ewing sarcoma, the scope of the work as presented is quite limited and the key mechanistic questions remain. Are SIX1 knockout cells viable? Tumorigenic in subcutaneous models? Do SIX1 and EWS::FLI1 cooperate at co-regulated and co-bound loci? Is this overlap limited to just one cell line? Is SIX1 expression heterogeneous in expression or activity - the introduction clearly articulates that this is of key importance for the fusion - is this also true for SIX1? In addition, some of the conclusions are premature and would require substantial additional support using more extensive experimentation and replication in additional models.

Major comments:

- The authors propose that "taken together, these data demonstrate that in the context of ES, SIX1 dramatically inhibits metastasis". Additional in vivo models (in more than one cell line) to support this claim. Tail vein injection only effectively models lung & liver outgrowth in Ewing sarcoma, not any of the initial stages of spontaneous metastasis. This could be a colonization phenotype. Without a subcutaneous or orthotopic primary tumor model it is unclear if SIX-KD cells merely have different proliferative/tumorigenic phenotypes in vivo vs. in vitro, rather than a metastasis-specific phenotype.
- The soft agar colony formation suggests a tumorigenic defect upon SIX1-KD that may be related or unrelated to its effects on migration.
- The authors should include gain-of-function experiments - e.g. if SIX1 actively suppresses migration/invasion/metastasis then overexpression of the protein in highly metastatic ES cells/tumor models would be expected to inhibit their metastatic properties.
- Combination studies that modulate SIX1 and EWS::FLI1 alone and together are needed to determine if SIX1 is functioning independently of the fusion and/or if they cooperate to repress metastasis-associated gene expression. Is this a property in ES cells alone or can it be modeled in other cells especially MSCs, the putative cell of origin of ES tumors?
- The finding that >80% of EWS::FLI1 bound genes are also bound by SIX1 is intriguing, but these experiments need to be repeated in additional cell lines, preferably under both normal and EWS::FLI1 knockdown conditions, to fully characterize SIX1 genome distribution and if/how it is influenced by the fusion
- Where does SIX1 bind in normal MSCs and does this change in the presence of EWS::FLI1 (or vice versa)?
- The finding that function-blocking antibodies directed at integrins reduces SIX-KD induced migration is intriguing. However, DMSO is an inadequate control. Appropriate diluent and IgG controls are needed
- While the potential role for SIX1-regulated integrins in metastatic migration of Ewing sarcoma is interesting, this requires additional functional assessment: ie. inhibition during in vivo metastasis assays; assessment of the impact of SIX1 on integrin-mediated signaling such as FAK or SRC;
- The claim that SIX1 is "predictive of event free survival" cannot be made without validation in more than one dataset. At most the curves show an association with improved EFS in one study of tumor specimens (this is also a prognostic not a predictive association). More study is required before any claims of prognostic significance can be made.

Reviewer #2:

Remarks to the Author:

The study by Hughes et al explores the role of Six1 in Ewing Sarcoma (ES). In doing so, the authors uncovered several interesting and unexpected findings related to Six1 action in Ewing Sarcomas (ES). Notably, the authors show that despite its ability to support an EMT program in ES cells, Six1 nevertheless functions as a suppressor of metastasis, doing so by repressing genetic programs coupled to the metastatic cascade. Importantly, these unusual Six1 activities parallel those of EWS:FLI1, with both proteins colocalizing on the promoters of genes essential to metastasis. Although potentially interesting and novel, the study itself is largely phenomenological and lacks sufficient mechanistic insights. Indeed, several major and important questions are never addressed, including (i) what mechanism(s) overrides the functionality of EMT programs in Six1-expressing ES cells; (ii) what is the nature and/or composition of Six1 repressive complexes; and (iii) what regulates Six1 protein stability in EWS-502 and other ES cells that express low levels of Six1 mRNA? Additionally, "cause/effect" analyses are generally underdeveloped. As such, attempts to address 1 or more of these issues is warranted to improve the impact and significance of the work. Additional comments and concerns are presented below under "Specific Comments."

Specific Comments:

1) The primary weakness of this potentially interesting study relates to the fact that Six1 drives and supports what amounts to a nonfunctional EMT program. This is an intriguing finding that needs additional validation and mechanistic support. Some attempts to monitor common EMT markers regulated by Six1 should be shown in parental and Six1-manipulated cells (A673 and EWS-502). Also, EMT programs elicit a spectrum of phenotypes, and as such, does chemoresistance, anti-apoptotic, or stemness remain functional in Six1-manipulated ES cells? Likewise, does loss of EWS:FLI1 also impact EMT programs downstream of Six1? In general, some attempts to establish how ES cells bypass the EMT program should be undertaken.

2) A second weakness of the study relates to the nature/composition of Six1 repressive complexes. Does this event reflect the actions of Six1-Dach complexes or other repressor proteins? Likewise, how are these events circumvented to enable Six1 to promote a non-functional EMT program in these same ES cells – e.g., does Eya mediate the conversion from repression to activation? Recently, HOXD13 was identified as a transcription factor capable of targeting and inducing the expression EWS:FLI1-repressed genes. Given the colocalization of EWS:FLI1 and Six1 at promoters, does binding of HOXD13 (or another transcription factor) modulate transcriptional programs governed by Six1? Some attempts to address this point is also warranted to elevate the impact of this unexpected finding.

3) Figure 1: What accounts for the dramatic stabilization of Six1 expression in EWS-502 (and other) cells that express low levels of Six1 mRNA? Does regulation of Six1 stabilization underlie the switch from metastasis-suppressing to metastasis-promoting? Some attempts to strengthen this aspect of the paper seems warranted.

4) Figure 3: Does manipulating Six1 expression in EWS-502 cells alter their metastatic competency? This data should be presented, as should some basic histological analyses.

5) Figure 6: It is unusual to find so many integrins upregulated in response to Six1-deficiency. Given that all Six1-regulated alpha integrins partner with beta1 integrin, it seems imperative to also deplete beta1 integrin in Six1-deficient ES cells to gauge its importance in driving metastasis of ES cells.

6) The "Discussion" section needs to be reworked. , as it simply restates the findings of the study as opposed to discussing the significance and impact of the work.

Reviewer #3:

Remarks to the Author:

The authors provide compelling data that the transcription factor SIX1 suppresses integrin subunit expression in Ewing's Sarcoma and it is this that explains the unexpected observation that high SIX1 correlates with a metastasis suppressive phenotype. In contrast in multiple carcinomas SIX1 is associated with promoting metastasis and correlating with poor Overall survival. In the manuscript a parallel to the Ewing Sarcoma fusion protein EWS/FLI1 behaviour is drawn in that over-expression of the fusion protein also is metastasis suppressive.

As written this is a fine study, well written and the data are convincing. This reviewer believes that the story remains unfinished and thus needs additional work.

1. Can authors provide STR sequencing proof that shRNA KD lines used in the paper are derived directly from the parental lines

2. The authors do not explore why SIX1 is metastasis suppressive in Ewing's and metastasis promoting in multiple carcinomas. Their explanation in Ewing's is that SIX1 binds to the promoter of multiple integrin subunit genes and suppresses their transcription. Does that happen in the several carcinomas where SIX1 drives metastasis. If it does then the hypothesis presented here is wrong. If SIX1 does not suppress integrins in carcinomas then explain why. At the end of the results the authors imply that SIX1 may function differently in Ewing's versus carcinomas because of the presence of the fusion protein EWS/FLI1. That is easily testable. Eliminate it in the Ewing's lines and introduce it into the carcinoma lines then re-examine the integrin regulation by SIX1.

3. If EWS/FLI1 is the regulator of the ability of SIX1 to regulate integrin genes then more effort is needed to determine how the two molecules can do this. Does EWS/FLI1 block part of the SIX1 binding site?

4. Please correctly refer to integrin proteins in the text and do not just use the gene names. Integrins are heterodimers so when describing the ligand binding abilities give the full integrin name i.e. $\alpha1\beta1$, $\alpha2\beta1$, $\alpha4\beta1$, $\alpha5\beta1$ and $\alpha6\beta1$.

5. There are no data on changes in surface expression of integrins. This is easily solved by doing flow cytometry for the several subunits mentioned. Integrins are not trypsin-sensitive so the data will be clean and should correspond with functional data.

6. The authors highlight that collagen is a major ligand for $\alpha1\beta1$ and $\alpha2\beta1$. Would the authors suggest a potential mechanism of reduced migration in the transwell assays in the presence of antibodies that blocked these integrins, given no collagen was present in the assay?

Please find enclosed our revised version of the manuscript “SIX1 and EWS/FLI1 co-regulate an anti-metastatic gene network in Ewing Sarcoma” by Hughes et al. We have completed all requested revisions, and believe that the addition of experiments suggested by reviewers has significantly improved this manuscript. We outline below our responses to the reviewers’ comments point by point. In the revised text, new/updated text and data are marked in blue.

Reviewer #1: Here the authors explore the role of the transcription factor SIX1 in Ewing sarcoma. The authors propose that SIX1 expression, in concert with EWS::FLI1 expression, exerts an anti-metastatic effect in Ewing sarcoma. This is an innovative discovery given that it is contrary to the role of SIX1 in most other cancers. SIX1-knock down reduced anchorage independent growth in vitro, but increased migration in 2D assays and outgrowth after tail vein injection. The authors identified a striking overlap in genes bound by both EWS::FLI1 and SIX1, and propose that a subset of these are co-regulated by both TFs to reduce metastatic potential, partly through repressing a subset of integrins. Integrin blocking antibodies reduced migration of SIX1 knock down cells in 2D revealing that repression integrins is at least partially responsible for the “anti-metastatic” properties of SIX1 in Ewing sarcoma cells. While these findings are intriguing and partially advance our understanding of pro- and anti-

metastatic gene networks in Ewing sarcoma, the scope of the work as presented is quite limited and the key mechanistic questions remain. Are SIX1 knockout cells viable? Tumorigenic in subcutaneous models? Do SIX1 and EWS::FLI1 cooperate at co-regulated and co-bound loci? Is this overlap limited to just one cell line? Is SIX1 expression heterogenous in expression or activity - the introduction clearly articulates that this is of key importance for the fusion- is this also true for SIX1? In addition, some of the conclusions are premature and would require substantial additional support using more extensive experimentation and replication in additional models.

All the questions in the above paragraph are addressed specifically below:

1. The authors propose that “taken together, these data demonstrate that in the context of ES, SIX1 dramatically inhibits metastasis”. Additional in vivo models (in more than one cell line) are needed to support this claim. Tail vein injection only effectively models lung & liver outgrowth in Ewing sarcoma, not any of the initial stages of spontaneous metastasis. This could be a colonization phenotype. Without a subcutaneous or orthotopic primary tumor model it is unclear if SIX-KD cells merely have different proliferative/tumorigenic phenotypes in vivo vs. in vitro, rather than a metastasis-specific phenotype.

Response: In response to the reviewer’s request to add additional models, we examined the role of SIX1 in regulating metastasis in a second *in vivo* tail vein model (EWS-502). EWS-502 cells with or without SIX1 KD had significantly reduced fitness *in vitro* when tagged with firefly

luciferase. As such, we were concerned this may impact survival and outgrowth *in vivo*. Thus, we performed tail-vein injections with untagged EWS-502 SCR and SIX1 KD cells. After growth for 33 days, the mice were sacrificed and dissected to look for evidence of metastasis. Macroscopic metastatic nodules were only observed on livers of the SIX1 KD groups, and not on any livers in the SIX1 control group. We fixed and performed H&E staining on all mouse livers to look for histological evidence of metastatic outgrowth. Representative images of H&E-stained liver sections from EWS-502 SCR and SIX1 KD cells demonstrate normal liver histology in mice injected with SCR cells and both concentric and diffuse patterns of infiltration in mice injected with SIX1 KD cells (Please see **Rebuttal Fig. 1**, now **revised Manuscript Figure 3D**, also inserted above). Of note, the EWS-502 model appears to have weak *in vivo* fitness overall, as few mice had detectable liver lesions. Nonetheless, all mice which grew metastatic lesions (as confirmed by histology) were mice injected with SIX1 KD cells (4/9). We observed no metastatic outgrowth in mice injected with EWS-502 SCR cells (0/5) (**Rebuttal Fig. 1**, **Revised Manuscript Fig. 3E**). These results show that the pro-metastatic effect of SIX1 KD occurs across ES models.

Rebuttal Fig. 2. (Revised Suppl. Figure 2). SIX1 KD attenuates subcutaneous tumor growth. (A) Caliper measurements of A673 SCR or SIX1 KD tumors after subcutaneous flank injection in NSG mice. A673 SCR (n=6), shSIX1 KD1 (n=5), and shSIX1 KD2 (n=4). Statistical analysis performed using a longitudinal mixed-effects model

We appreciate the reviewer's insight into a potential caveat to the tail-vein injection model regarding cell growth. To determine whether SIX1 KD ES cells have differences in growth *in vivo*, a subcutaneous model was performed, as suggested. A673 SCR or SIX1 KD cells were subcutaneously injected into NSG mice to look for SIX1-regulated effects on tumorigenicity/proliferation *in vivo*. In this model, SIX1 KD led to a significant *reduction* in subcutaneous tumor growth of A673 cells (**Rebuttal Fig. 2**, **Revised Suppl. Fig. 2A**), in line with the *in vitro* effects of A673 growth in soft agar (Fig. 1I). This finding demonstrates that SIX1 appears to directly repress the metastatic propensity of ES cells even while promoting cell growth/tumorigenicity. Of note, we also performed subcutaneous flank injections with EWS-502 SCR and SIX1 KD cells, however after 35 days we did not observe any outgrowth of tumors in either group, in line with the observation of EWS-502 cells as a relatively poor *in vivo* model of ES.

2. The soft agar colony formation suggests a tumorigenic defect upon SIX1-KD that may be related or unrelated to its effects on migration.

Response: We agree with the reviewer that the inhibition of cancer cell outgrowth in our soft agar colony formation assay upon SIX1 KD may be suggestive of a tumorigenic defect with SIX1 KD. As demonstrated in our new figure (**Rebuttal Fig. 2**, **Revised Suppl. Fig 2A**), A673 SCR cells are indeed more tumorigenic than SIX1 KD cells, in line with the soft agar colony formation assays seen in Fig. 1I-L in the manuscript. However, we observed more, not less, *in vitro* migration/invasion (Fig. 2A-H of the manuscript) and more *in vivo* metastatic outgrowth after tail vein injection (**Rebuttal Fig. 1**, **Revised Fig. 3A-E**) upon SIX1 KD, demonstrating that SIX1 has opposing effects on tumorigenicity and motility/metastasis in ES cells; similar to what has previously been observed with EWS-FLI1^{1,2}.

3. The authors should include gain-of-function experiments - e.g. if SIX1 actively suppresses migration/invasion/metastasis then overexpression of the protein in highly metastatic ES cells/tumor models would be expected to inhibit their metastatic properties.

Response: In our studies, we see very low migration/invasion and metastasis in both our models of ES with endogenous SIX1 expression, leaving little room to observe even higher repression of these phenotypes with gain-of-function experiments in these systems. The data presented in this manuscript suggest that endogenous SIX1 levels are already strongly anti-metastatic in the context of ES cells, and thus addition of ectopic SIX1 in this context (and in other ES lines, most of which already express SIX1 endogenously, see Fig. 1B of the revised manuscript) would unlikely yield significant effects. Thus, we feel the SIX1 KD approaches, now used in two different ES models, are best able to assess the function of SIX1 on migration, invasion, and metastasis.

4. Combination studies that modulate SIX1 and EWS::FLI1 alone and together are needed to determine if SIX1 is functioning independently of the fusion and/or if they cooperate to repress metastasis-associated gene expression. Is this a property in ES cells alone or can it be modeled in other cells especially MSCs, the putative cell of origin of ES tumors?

Response: To address the relative impact of SIX1 and EWS/FLI1 loss and whether the two proteins are acting cooperatively to impact metastasis-associated gene expression, we used an siRNA targeting the fusion sequence of EWS/FLI1³ to transiently reduce EWS/FLI1 expression in A673 SCR and shSIX1 KD cells. We then performed qRT-PCR looking at the expression of integrins genes that we demonstrated are regulated by SIX1. shRNA-mediated KD of SIX1, and siRNA-mediated KD of EWS-FLI1, was confirmed at the mRNA level (**Rebuttal Fig. 3., Revised Suppl. Figs. 13A, B**). In line with our prior findings, we consistently observe that SIX1 KD leads to significant increases in integrin gene expression. For three of the integrins we queried, ITGA1, ITGA6, and ITGB1, independent KD of EWS/FLI1 led to similar levels of enhanced gene expression as what was observed with SIX1 KD alone (**Rebuttal Fig. 3, Revised Suppl. Figs. 13C-E**). Combined KD of SIX1 and EWS/FLI1 led to similar or modest additional increased expression of the integrins over either KD alone, demonstrating that both proteins are needed to repress these integrins, highly suggestive of cooperativity at these loci (**Rebuttal Fig. 3, Revised Suppl. Figs. 13C-E**).

For other integrins, the combined effect of SIX1 and EWS-FLI1 loss did not appear cooperative, and may in fact be antagonistic. For example, for ITGA4 and ITGA5, SIX1 KD led to a large increase in expression of both integrins, but there was a consistent reduction in the expression of these integrins with EWS/FLI1 KD, suggesting that EWS/FLI1 expression may actually maintain the expression of these two integrins when SIX1 is lost (**Rebuttal Fig. 3, Revised Suppl. Figs. 13F, G**). This finding suggests that SIX1 plays a

dominant repressive role across all queried integrins, with EWS/FLI1 expression enabling specific control over the expression of subsets of these integrins. These divergent patterns of regulation suggest a complex interplay between SIX1 and EWS/FLI1, yet clearly underscore cooperative and potentially competitive actions for the two proteins at specific gene loci. These findings open up several new interesting lines of inquiry that are beyond the scope of this manuscript, but remain future areas of investigation for our lab to better understand the exact complementary and opposing regulatory functions of SIX1 and EWS/FLI1 in the context of ES.

5. The finding that >80% of EWS::FLI1 bound genes are also bound by SIX1 is intriguing, but these experiments need to be repeated in additional cell lines, preferably under both normal and EWS::FLI1 knockdown conditions, to fully characterize SIX1 genome distribution and if/how it is influenced by the fusion

Response: In response to this concern, we performed CUT&RUN for FLI1, SIX1, and H3K27me3 in EWS-502 SCR and shSIX1 KD cells (to complement the CUT&RUN shown in the A673 system in the original version of the manuscript). We first looked at whether we observed enrichment of both SIX1 and FLI1 at the promoters of integrins observed in the A673 system. Both FLI1 and SIX1 peaks were once again observed at integrin

Rebuttal Fig. 4. (Revised Suppl. Fig. 9). SIX1 and FLI1 bind at integrin gene promoters in A673 and EWS-502 cells. CUT&RUN tracks for A673 SCR and shSIX1 KD cells around the promoter region of (A) ITGA5 and (B) ITGB1. Tracks shown are H3K4me1, H3K4me3, FLI1, and SIX1. CUT&RUN tracks for EWS-502 SCR and shSIX1 KD cells around the promoter region of (C) ITGA1, (D) ITGA2, (E) ITGA4, and (F) ITGB1. Tracks shown are FLI1, SIX1, and H3K27me3.

promoters in the EWS-502 system (**Rebuttal Fig. 4, Revised Supp. Figs. 9C-F**). We also looked for changes in the repressive mark H3K27me3 in this system, to see whether we see evidence of a repressive signature in the SCR cells relative to SIX1 KD cells. As observed in **Rebuttal Fig. 4 (Revised Supp. Figs. 9C-F)**, there is a general trend of increased H3K27me3 repressive marks on integrin promoters in EWS-502 SCR cells, with loss of these repressive marks occurring in SIX1 KD cells, in line with the downstream impact on expression. These data suggest a relationship between SIX1 expression and epigenetic repression of pro-metastatic genes.

In line with what was observed in the A673 system, we observed significant overlap in genes bound by both SIX1 and EWS/FLI1 (**Rebuttal Fig. 5, Revised Supp. Fig. 10A**) in the EWS-502 system. We additionally saw significant overlap of co-bound genes with DEGs upon SIX1 KD in the EWS-502 system, suggesting SIX1 is

regulating a significant number of co-bound genes (**Rebuttal Fig 5, Revised Supp. Fig. 10B**). We also looked at the overlap with EWS/FLI1 KD from a publicly available dataset in the A673 system⁴, and observed significant overlap with SIX1 DEGs in the EWS-502 system, demonstrating co-regulated genes across systems, many of which are co-bound by SIX1 and EWS/FLI1 (**Rebuttal Fig. 5, Revised Supp. Fig. 10C**). These genes enriched for adhesion associated gene sets (**Rebuttal Fig. 5, Revised Supp. Fig. 10D**), but not as many metastasis-associated pathways as in the A673 system (See manuscript Fig. 8G), but this may be a product of comparing overlapping genes across different systems. Of note, in line with the A673 system, most co-bound genes were promoter localized (**Rebuttal Fig. 6, Revised Supp. Fig. 11C, D – shown below**), suggesting that these are direct regulatory targets of EWS/FLI1 and SIX1. These findings demonstrate widespread overlap between SIX1 and EWS/FLI1 co-bound and co-regulated genes in multiple systems of ES, suggesting a conserved co-regulatory network between SIX1 and EWS/FLI1 in this context.

Due to the profound overlap of SIX1 and EWS/FLI1 bound genes in both the A673 and EWS-502 systems, we were interested in whether SIX1 and EWS/FLI1 were binding similar genetic motifs, as a means to explain why the two proteins exhibit such dramatic co-

traditionally characterized by variations of the purine-rich GGAA/T motif⁵ in both the A673 and EWS-502 systems. Of note, the FLI1 or EWS/FLI1 fusion binding motif was enriched at sites bound by both SIX1 and FLI1 in both the A673 (Rebuttal Tables 1, 2, Revised Supp. Tables 1, 2) and EWS-502 systems (Rebuttal Tables 3, 4, Revised Supp. Tables 3, 4), suggesting that SIX1 is binding at GGAA/T-enriched sequences genome-wide in this context. These data offer a potential explanation for the widespread overlap in genome-wide localization between SIX1 and EWS/FLI1, and demonstrate that the shared phenotypes observed with SIX1 or EWS/FLI1 loss are likely a product of co-localization and coordinate regulation of ETS TF family target genes.

Table 1. A673 FLI1 CUT and RUN Motif Analysis

Rank	Motif	Name	p-value	log P-value	q-value (Benjamini)	# Target Sequences with Motif	% of Targets Sequences with Motif
1		ETS(ETS) Promoter Homer	1e-137	-3.161e+02	0.0000	1472	24.05%
2		EB1(ETS) Hela-EB1-CMP-Seq(GSE31477) Homer	1e-121	-2.795e+02	0.0000	2192	35.82%
3		EB4(ETS) Hela-EB4-CMP-Seq(GSE31477) Homer	1e-118	-2.738e+02	0.0000	2200	35.95%
4		ELF1(ETS) Jukat-ELF1-CMP-Seq(SRA014231) Homer	1e-105	-2.419e+02	0.0000	1999	32.66%
5		EWS-FLI1 fusion(ETS) SK_N_MC-EWS-FLI1-CMP-Seq(SRA014231) Homer	1e-100	-2.310e+02	0.0000	1365	22.30%
6		HLI(ETS) CDR-FLI-CMP-Seq(GSE20898) Homer	1e-98	-2.264e+02	0.0000	2635	43.06%
7		GABPA(ETS) Jukat-GABPa-CMP-Seq(GSE17954) Homer	1e-97	-2.234e+02	0.0000	2180	35.13%
8		ETS(ETS) Jukat-ETS1-CMP-Seq(GSE17954) Homer	1e-79	-1.842e+02	0.0000	2119	34.62%
9		ETV4(ETS) ES-ETV4-CMP-Seq(GSE59422) Homer	1e-75	-1.731e+02	0.0000	1752	28.63%
10		ETV4(ETS) HepG2-ETV4-CMP-Seq(ENCODE) Homer	1e-74	-1.718e+02	0.0000	2770	45.26%

Table 2. A673 SIX1 CUT and RUN Motif Analysis

Rank	Motif	Name	p-value	log P-value	q-value (Benjamini)	# Target Sequences with Motif	% of Targets Sequences with Motif
1		EB4(ETS) Hela-EB4-CMP-Seq(GSE31477) Homer	1e-48	-1.120e+02	0.0000	3755	33.83%
2		EB1(ETS) Hela-EB1-CMP-Seq(GSE31477) Homer	1e-41	-9.629e+01	0.0000	3702	33.35%
3		YY1(ZD) Promoter Homer	1e-41	-9.496e+01	0.0000	762	6.86%
4		ETS(ETS) Promoter Homer	1e-36	-8.475e+01	0.0000	2145	19.32%
5		ELF1(ETS) Jukat-ELF1-CMP-Seq(SRA014231) Homer	1e-35	-8.067e+01	0.0000	3354	30.22%
6		TFE3(MLH) MEF-TFE3-CMP-Seq(GSE3757) Homer	1e-28	-6.643e+01	0.0000	466	4.20%
7		E-box(MLH) Promoter Homer	1e-27	-6.222e+01	0.0000	667	6.01%
8		HLI(ETS) CDR-FLI-CMP-Seq(GSE20898) Homer	1e-26	-6.202e+01	0.0000	4573	41.20%
9		zinc1(TIAP) ES-Thap11-CMP-Seq(GSE3522) Homer	1e-21	-4.942e+01	0.0000	445	4.01%
10		NYF(CCAAT) Promoter Homer	1e-20	-4.736e+01	0.0000	2235	20.14%

localization throughout the genome. Thus, we performed motif enrichment analysis on both the A673 SCR and EWS-502 SCR CUT&RUN peaks for FLI1 and SIX1 to determine which DNA binding motifs could be identified in both systems. Strikingly, we observed strong enrichment for many ETS family transcription factor motifs for both FLI1 and SIX1 (Rebuttal Table 1-4, Revised Supp. Tables 1-4 – truncated tables included below, see revised manuscript for full tables),

Table 3. EWS-502 FLI1 CUT and RUN Motif Analysis

Rank	Motif	Name	p-value	log P-value	q-value (Benjamini)	# Target Sequences with Motif	% of Targets Sequences with Motif
1		EB4(ETS) Hela-EB4-CMP-Seq(GSE31477) Homer	1e-188	-4.343e+02	0.0000	2318	23.87%
2		EB1(ETS) Hela-EB1-CMP-Seq(GSE31477) Homer	1e-188	-4.340e+02	0.0000	2318	23.87%
3		HLI(ETS) CDR-FLI-CMP-Seq(SRA014231) Homer	1e-164	-3.789e+02	0.0000	2788	28.20%
4		ELF1(ETS) Jukat-ELF1-CMP-Seq(SRA014231) Homer	1e-153	-3.538e+02	0.0000	2057	21.19%
5		ETV4(ETS) HepG2-ETV4-CMP-Seq(ENCODE) Homer	1e-151	-3.481e+02	0.0000	2954	30.43%
6		GABPA(ETS) Jukat-GABPa-CMP-Seq(GSE17954) Homer	1e-148	-3.411e+02	0.0000	2145	22.09%
7		ETV1(ETS) GIST48-ETV1-CMP-Seq(GSE22414) Homer	1e-147	-3.391e+02	0.0000	2695	27.76%
8		ETS(ETS) Promoter Homer	1e-144	-3.317e+02	0.0000	1414	14.56%
9		ETS(ETS) Jukat-ETS1-CMP-Seq(GSE17954) Homer	1e-112	-2.595e+02	0.0000	2050	21.11%
10		EB4(ETS) BMDM-EB4-CMP-Seq(GSE38869) Homer	1e-110	-2.542e+02	0.0000	1929	19.87%

Table 4. EWS-502 SIX1 CUT and RUN Motif Analysis

Rank	Motif	Name	p-value	log P-value	q-value (Benjamini)	# Target Sequences with Motif	% of Targets Sequences with Motif
1		NRF(NRF) Promoter Homer	1e-114	-2.628e+02	0.0000	1442	23.44%
2		NYF(CCAAT) Promoter Homer	1e-97	-2.244e+02	0.0000	1499	24.37%
3		Sp1(ZD) Promoter Homer	1e-96	-2.226e+02	0.0000	2660	43.25%
4		NRF(NRF) MCF7-NRF1-CMP-Seq(Unpublished) Homer	1e-93	-2.152e+02	0.0000	1507	24.50%
5		Sp5(ZD) mES-Sp5-Flag-CMP-Seq(GSE72989) Homer	1e-70	-1.615e+02	0.0000	4063	66.05%
6		KLF3(ZD) MEF-KLF3-CMP-Seq(GSE4748) Homer	1e-68	-1.572e+02	0.0000	2505	40.73%
7		KLF1(ZD) HUDEF2-KLF1-CatBun(GSE136251) Homer	1e-65	-1.511e+02	0.0000	3623	58.90%
8		ELF1(ETS) Jukat-ELF1-CMP-Seq(SRA014231) Homer	1e-56	-1.299e+02	0.0000	1778	28.91%
9		KIP2(ZD) GBM-KIP2-CMP-Seq(GSE62211) Homer	1e-56	-1.299e+02	0.0000	1971	32.04%
10		EB4(ETS) Hela-EB4-CMP-Seq(GSE31477) Homer	1e-55	-1.272e+02	0.0000	1935	31.46%

Rebuttal Tables 1-4. (Revised Suppl. Tables 1-4 - truncated). SIX1 and FLI1 bind at ETS Transcription Factor Motifs. Top enriched motifs with known motif analysis based on (A) A673 SCR FLI1 and (B) A673 SCR SIX1 CUT&RUN tracks. Top enriched motifs with known motif analysis based on (C) EWS-502 SCR FLI1 and (D) EWS-502 SCR SIX1 CUT&RUN tracks.

Rebuttal Fig. 7. (Revised Suppl. Fig. 8). SIX1 does not regulate EWS/FLI1 expression but EWS/FLI1 enhances SIX1 protein levels Western blotting analysis for of SIX1 and FLI1 in (A) A673 SCR and SIX1 KD cells and (B) EWS-502 SCR and SIX1 KD cells. qRT-PCR analysis for (C) EWS/FLI1 expression and (D) SIX1 expression in A673 A1C (dox-inducible EWS/FLI1 KD) cells treated with the indicated time course of 2 μ g/ml doxycycline. Mean \pm SD of technical replicates shown. Statistical analysis performed with a one-way ANOVA with post-hoc Dunnett's multiple comparisons test. (E) Western blotting analysis for FLI1 and SIX1 in time course of doxycycline treatment in A673 A1C cells. (F) Western blotting analysis for SIX1 levels in A673 SCR cell with 48h treatment with 2 μ g/ml doxycycline as a control for E.

6. Where does SIX1 bind in normal MSCs and does this change in the presence of EWS::FLI1 (or vice versa)?

Response: As seen in Fig. 1B of the revised manuscript, SIX1 expression is nearly absent in hMSCs, and thus we felt that examining its binding through CUT&RUN in this setting may not be a fruitful approach to take. Of note, subsequent experimentation performed since our initial submission determined that SIX1 protein expression is enhanced in the presence of EWS/FLI1 (**Rebuttal Fig. 7, Revised Suppl. Fig. 8**)(**Rebuttal Fig. 8, Revised Manuscript Fig. 7**), indicating that it may be the case that expression of SIX1 is induced/stabilized by EWS/FLI1 in ES, and we would likely not have sufficient SIX1 expression in hMSCs without ectopic EWS/FLI1 expression in order to determine genome-wide binding patterns of SIX1 in the absence of EWS/FLI1. As a consequence, we have not compared SIX1 genomic localization with or without EWS/FLI1 expression due to low baseline SIX1 expression in hMSCs.

7. The finding that function-blocking antibodies directed at integrins reduces SIX-KD induced migration is intriguing. However, DMSO is an inadequate control. Appropriate diluent and IgG controls are needed

Response: We appreciate the reviewer's attention to detail in catching this incorrect control. We have repeated these assays with the appropriate IgG control (**Rebuttal Fig. 9, Revised Manuscript Figure 6G-J – shown below**). Use of the appropriate control did not alter the outcome, as SIX1 KD still enhanced invasion of ES cells in a manner dependent on integrin activation when this control was included. Of note, we also added an integrin β 1-neutralizing antibody, as this is a shared heterodimeric partner of all alpha integrins queried, and therefore will likely block several different integrin sub-species at once. As a clarifying point, these assays were performed with Matrigel-coated transwells, and were thus invasion assays.

Rebuttal Fig. 8. (Revised Manuscript Fig. 7). EWS/FLI1 regulates SIX1 protein levels but not transcription. Representative qRT-PCR expression of (A) EWS/FLI1 and (B) SIX1 in A673 SCR and shFLI1 KD cells. qRT-PCR expression of (C) EWS/FLI1 and (D) SIX1 in EWS-502 SCR and shFLI1 KD cells. For all qRT-PCR experiments, Mean \pm SD of technical replicates shown. Statistics performed using one-way ANOVA with post-hoc Dunnett's multiple comparisons test. (E) Western blotting analysis of A673 SCR and shFLI1 KD cells for SIX1 levels. (F) Western blotting analysis of EWS-502 SCR and shFLI1 KD cells for SIX1 levels.

8. While the potential role for SIX1-regulated integrins in metastatic migration of Ewing sarcoma is interesting, this requires additional functional assessment: ie. inhibition during in vivo metastasis assays; assessment of the impact of SIX1 on integrin-mediated signaling such as FAK or SRC.

Response: To assess whether SIX1-mediated integrin repression was impacting not only surface

expression of integrins, but also signaling downstream of integrin activation, we performed Western blot analysis on A673 and EWS-502 SCR and SIX1 KD cells and looked for alterations in the activating phospho-marks p-FAK (Y379) and p-SRC (Y416) upon SIX1 KD. As seen in **Rebuttal Fig. 10, Revised Manuscript Figs. 5E, F**, SIX1 KD leads to robust increases in p-FAK and p-SRC levels, demonstrating that SIX1-mediated integrin repression is driving reductions not only in integrin expression, but also in downstream intracellular signaling through integrin proteins.

To confirm that the increased integrin expression observed in SIX1 KD cells is promoting the increased metastatic outgrowth seen in both the A673 and EWS-502 systems, we attempted to use shRNA targeting integrin β 1 to stably KD the shared partner of all alpha integrins studied in this manuscript.

Unfortunately, despite repeated attempts, the SIX1 KD cells could not tolerate KD of integrin β 1, suggesting that upregulation of integrin β 1 is important for survival of these cells upon SIX1 loss. As such, we felt alternative means of integrin neutralization *in vivo*, such as antibody-mediated blocking, could have confounding results, as neutralization may cause death of the SIX1 KD cells independent

Rebuttal Fig. 10. (Revised Manuscript Figure 5C-F). SIX1 KD Increases membrane expression and downstream activation of integrins. Western blot analysis of membrane-specific expression of the aforementioned integrins between SCR and SIX1 KD cells in (C) A673 and (D) EWS-502 cell lines. Representative GAPDH shown as a membrane purity control and representative NaKATPase as a membrane protein loading control. Western blot analysis of activated p-FAK (Y379) and p-SRC (Y416) in SCR and SIX1 KD cells in the (E) A673 and (F) EWS-502 cell lines. Representative SIX1 and β -Actin blots shown.

of a metastasis-specific phenotype. As there are widespread integrin changes upon SIX1 KD, blocking of individual alpha integrins may not be sufficient to fully prevent metastatic outgrowth of SIX1 KD cells, and the impact would likely depend on which integrin was targeted. Given our data demonstrating SIX1 KD not only increases membrane integrin expression (Revised Manuscript Fig. 5C, D) but also downstream signaling (Revised Manuscript Fig. 5E, F), and neutralization of multiple integrins *in vitro* reduces invasion of SIX1 KD cells in both the A673 and EWS-502 systems (Revised Manuscript Figs. 6G-J), we feel this is sufficient to argue that integrin repression is likely an important mechanism of metastasis suppression downstream of SIX1 in ES. This statement is taken within the context of a breadth of prior literature demonstrating direct roles of integrins in promoting metastasis across a variety of tumor types⁶⁻¹⁰, including sarcomas^{11,12}. However, we have been careful within the manuscript to not state that we have demonstrated *in vivo* that metastasis downstream of SIX1 is suppressed through integrin inhibition, but rather that the effects of SIX1 on integrins likely contributes to its anti-metastatic effect *in vivo*.

9. The claim that SIX1 is “predictive of event free survival” cannot be made without validation in more than one dataset. At most the curves show an association with improved EFS in one study of tumor specimens (this is also a prognostic not a predictive association). More study is required before any claims of prognostic significance can be made.

Response: We appreciate the reviewer’s observation that the human data analysis we performed can demonstrate only a correlation rather than a predictive association – we have updated the text to reflect the limitations of using a single study, and to use language that demonstrates correlation and not causality. We agree that analysis by one dataset limits the generalizability of these claims, however, to our knowledge there are only two published datasets of ES patients which have both expression and survival data (likely due to the low prevalence of the disease). Volchenboum et al. 2015¹³ (GEO ID GSE63157) was used for our analysis, due to the higher sample size in this dataset (n=85). To address the reviewer’s

concern we also performed EFS survival analysis on Savola et al. 2011¹⁴ (GEO ID GSE17679) (**Rebuttal Fig. 11, Response Fig. 1A**). Unfortunately, due to low sample size (n=44, the dataset has duplicates of each patient), and due to the fact that 10 tumors in this dataset were characterized as Primitive Neuro-Ectodermal Tumors (PNETs) rather than ES (these were excluded), it is difficult to draw firm conclusions from these results. Nonetheless, there is a general trend towards higher SIX1 conferring longer EFS in this dataset when the top tercile of SIX1 expression is compared to the bottom tercile. Because this sample size is small, we did not feel it appropriate to include the analysis of this dataset in the manuscript. Nonetheless, we agree with the limitations of performing survival analyses on only one dataset, and therefore we have updated the manuscript text to convey this limitation.

Reviewer #2: The study by Hughes et al explores the role of Six1 in Ewing Sarcoma (ES). In doing so, the authors uncovered several interesting and unexpected findings related to Six1 action in Ewing Sarcomas (ES). Notably, the authors show that despite its ability to support an EMT program in ES cells, Six1 nevertheless functions as a suppressor of metastasis, doing so by repressing genetic programs coupled to the metastatic cascade. Importantly, these unusual Six1 activities parallel those of EWS:FLI1, with both proteins colocalizing on the promoters of genes essential to metastasis. Although potentially interesting and novel, the study itself is largely phenomenological and lacks sufficient mechanistic insights. Indeed, several major and important questions are never addressed, including (i) what mechanism(s) overrides the functionality of EMT programs in Six1-expressing ES cells; (ii) what is the nature and/or composition of Six1 repressive complexes; and (iii) what regulates Six1 protein stability in EWS-502 and other ES cells that express low levels of Six1 mRNA? Additionally, “cause/effect” analyses are generally underdeveloped. As such, attempts to address 1 or more of these issues is warranted to improve the impact and significance of the work. Additional comments and concerns are presented below under “Specific Comments.”

Comments in the above paragraph are addressed below in specific comments.

1. The primary weakness of this potentially interesting study relates to the fact that Six1 drives and supports what amounts to a nonfunctional EMT program. This is an intriguing finding that needs additional validation and mechanistic support. Some attempts to monitor common EMT markers regulated by Six1 should be shown in parental and Six1-manipulated cells (A673 and EWS-502). Also, EMT programs elicit a spectrum of phenotypes, and as such, does chemoresistance, anti-apoptotic, or stemness remain functional in Six1-manipulated ES cells? Likewise, does loss of EWS:FLI1 also impact EMT programs downstream of Six1? In general, some attempts to establish how ES cells bypass the EMT program should be undertaken.

Response: We appreciate the reviewer’s comments related to our unique observations related to EMT downstream of SIX1 in this context, an area in which we are also very interested. As observed in Fig. 4 within the manuscript, the Hallmark Epithelial Mesenchymal Transition (EMT) gene set was predominantly downregulated upon SIX1 KD in our ES cell lines, however as noted by the reviewer, this did not lead to the typical outcome of reducing metastasis and metastatic phenotypes upon SIX1 loss. To understand the relationship between SIX1, EMT, and phenotypes associated with a traditional EMT in this context, we performed a series of additional experiments. We first performed qRT-PCR on a panel of EMT transcription factors and biomarkers shown to be regulated by SIX1 in epithelial cells and carcinomas¹⁵⁻¹⁹ to determine if

these may be differentially regulated by SIX1 in ES cells. To our surprise, many of these markers show differential effects in response to SIX1 KD in ES cells relative to other tumor and cell types^{16,17,20}. For example, loss of SIX1 reduced gene expression of both CDH1 (E-cadherin, an epithelial marker) and CDH2 (N-cadherin, a mesenchymal marker) in both the A673 and EWS-502 systems (**Rebuttal Fig. 12, Revised Supp. Figs. 3A, B, G, H**), whereas in epithelial tumors loss of SIX1 leads to upregulation of CDH1 and downregulation of CDH2^{16,17,21}. Most mesenchymal markers are downregulated by SIX1 KD in ES (similar to what is observed in epithelial tumors), though in EWS-502 cells, FN is upregulated with SIX1 KD (**Rebuttal Fig. 12, Revised Supp. Fig. 3L**). These data suggest more mixed effects of SIX1 on EMT genes in ES. A similar phenomenon is observed when examining the expression of EMT-related transcription factors (TFs), SNAI1 and TWIST1 (**Rebuttal Fig. 12, Revised Supp. Figs. 3C, D, I, J**) in the context of ES. While SNAI1 levels are reduced, as anticipated, by SIX1 KD in both A673 and EWS-502 cells, TWIST1 is downregulated in one context but upregulated in the other. These data suggest that in the context of ES, SIX1 can regulate EMT associated genes, but that the regulation is not uniformly pro-mesenchymal. These data are not surprising in the context of sarcomas, where EMT and EMT-associated phenotypes are poorly defined given the mesenchymal features of the tumors *ab initio*.²²

To further gauge how these changes in gene expression relate to functional phenotypes associated with EMT, we performed a series of experiments, including flow cytometric analysis to assess apoptosis via propidium iodide staining and chemotherapeutic resistance assays to multiple standard of care chemotherapeutics used to treat ES²³. SIX1 KD had no consistent effect on apoptosis in the A673 system, however in the EWS-502 system, SIX1 KD actually appeared to reduce the apoptotic rate relative to SCR cells (**Rebuttal Fig. 13, Revised Supp. Figs. 4A, B**) again in contrast to previously observed effects on apoptosis with SIX1 loss²⁴⁻²⁶. We further examined sensitivity of ES cells with or without SIX1 KD to the chemotherapeutic agents Vincristine and Actinomycin D, both of which are standard of care treatments for ES tumors²³. While SIX1 KD has no effect on Vincristine sensitivity in either system (**Rebuttal Fig. 13, Revised Supp. Figs. 4C, D**), it did increase the sensitivity of both A673 and EWS-502 cells to Actinomycin D (**Rebuttal Fig. 13, Revised Supp. Figs. 4E, F**), in line with the canonical role of EMT in promoting chemoresistance. These results demonstrate that while SIX1 KD still is associated with reduction in an EMT transcriptional signature (Fig. 4A-D in manuscript), SIX1 regulates a mixed EMT phenotype in ES cells. We speculate that by creating this mixed phenotype, loss of SIX1 may result in a cellular phenotype that is more “undifferentiated” and possibly more plastic, which may in part contribute to the fact that SIX1 loss makes cells more metastatic in this context. Indeed, EMT in tumors of mesenchymal origin is known to have more nuanced effects, often producing a more plastic or “metastable” cellular phenotype as opposed to promoting a canonical EMT as observed in carcinomas²². Finally, our

findings suggest that SIX1 may play a more specific role in mediating resistance to classes of chemotherapeutic compounds which target transcription, rather than causing generalized chemoresistance, which is often seen with a canonical EMT²⁷. In sum, these findings argue that SIX1-associated gene expression promotes a non-canonical or mixed EMT in ES. Additional commentary on this has been added to the Discussion section of the manuscript.

2. A second weakness of the study relates to the nature/composition of Six1 repressive complexes. Does this event reflect the actions of Six1-Dach complexes or other repressor proteins? Likewise, how are these events circumvented to enable Six1 to promote a non-functional EMT program in these same ES cells – e.g., does Eya mediate the conversion from repression to activation? Recently, HOXD13 was identified as a transcription factor capable of targeting and inducing the expression of EWS:FLI1-repressed genes. Given the colocalization of EWS:FLI1 and Six1 at promoters, does binding of HOXD13 (or another transcription factor) modulate transcriptional programs governed by Six1? Some attempts to address this point is also warranted to elevate the impact of this unexpected finding.

Response: To determine if SIX1 is driving gene repression in this context through forming a repressive complex with DACH1, we performed co-immunoprecipitation assays to look for binding of SIX1 and DACH1 in the A673 system (**Rebuttal Fig. 14, Response Fig. 2**). Based on these findings, there does not appear to be any direct interaction between DACH1 and SIX1 in this context, suggesting a unique mechanism of gene repression in ES cells. EYA proteins have only been characterized as co-activators of gene transcription; no co-repressive role has previously been demonstrated^{21,28}. Thus, we did not directly examine the EYA proteins in this context.

We appreciate the reviewer's observation of the recent study by Apfelbaum et al. 2022²⁹ which demonstrates quite elegantly that HOXD13 can induce the expression of EWS/FLI1-repressed genes, and there are certainly interesting parallels given the functions of both HOXD13 and SIX1 as developmental transcription factors. We did not see any changes in HOXD13 protein levels with SIX1 KD in either the A673 or EWS-502 systems (**Rebuttal Fig. 15, Response Fig. 3A, B** – shown above), indicating that it is likely that SIX1 is not directly regulating HOXD13 protein expression to regulate EWS/FLI-associated gene expression. Alternatively, it is possible that SIX1 is not regulating HOXD13 expression directly, but rather is binding at similar genetic loci to HOXD13, thereby leading to direct competition for regulation of shared genetic targets.

As such, we compared the genetic localization of SIX1 and HOXD13 using our A673 SIX1 CUT&RUN compared to A673 HOXD13 CUT&RUN published by Apfelbaum et al. 2022²⁹. HOXD13 primarily localized to intronic and intergenic regions, suggesting indirect means of regulating target genes (**Rebuttal Fig. 15, Response Figs. 3C, D**). Alternatively, SIX1 was primarily localized at the promoters of genetic targets (**Rebuttal Fig. 15, Response Figs. 3C, D**),

suggesting that these two proteins likely regulate the expression of EWS/FLI1-target genes through independent mechanisms. Nonetheless, the similarities in these regulatory patterns are noted, and suggest a propensity for developmental transcription factors to play a role in modulating EWS/FLI1 activity in ES, something to examine more carefully in future work.

- Figure 1: What accounts for the dramatic stabilization of Six1 expression in EWS-502 (and other) cells that express low levels of Six1 mRNA? Does regulation of Six1 stabilization underlie the switch from metastasis-suppressing to metastasis-promoting? Some attempts to strengthen this aspect of the paper seems warranted.

Response: As seen in Figure 1 of our manuscript, across our panel of ES cell lines, mRNA levels of SIX1 generally correlate with protein expression levels, however this is not true of mesenchymal stem cells, the presumed cell of origin for ES. In line with its function as a developmental transcription factor, SIX1 is often reduced in expression after development, and then in many contexts re-expressed in transformed cells¹⁶. Nonetheless, the reviewer poses an interesting question as to whether EWS/FLI1 can stabilize the SIX1 protein. In our initial paper, we showed that transient doxycycline-inducible EWS/FLI1 KD in the A673 system did not lead to any appreciable reduction in SIX1 protein levels. However, to more thoroughly address this question, we performed time-course experiments to determine whether a more stable EWS-FLI1 KD could affect SIX1 levels. While KD of SIX1 does not affect EWS/FLI1 protein levels (**Rebuttal Fig. 16, Revised Supp. Figs. 8A, B**), KD of EWS-FLI1 over time does lead to a reduction in SIX1 protein levels, without altering mRNA levels (**Rebuttal Fig. 16, Revised Supp. Fig. 8C-E**). To further validate this finding, we performed stable shRNA-mediated EWS/FLI1 KD in both cell lines (**Rebuttal Fig. 17, Revised Manuscript Fig. 7E, F**). These data indicate that EWS/FLI1 is promoting SIX1 expression at a post-transcriptional level, and this may explain the incongruence between SIX1 mRNA expression and protein expression in ES lines vs. hMSCs (Figs. 1A, B). As such, EWS/FLI1-low cells may have higher expression of pro-metastatic genes, at least in part, as a consequence of having less SIX1 protein expression, thereby de-repressing SIX1 bound genes.

- Figure 3: Does manipulating Six1 expression in EWS-502 cells alter their metastatic

Rebuttal Fig. 16. (Revised Suppl. Fig. 8). SIX1 does not regulate EWS/FLI1 expression but EWS/FLI1 enhances SIX1 protein levels
Western blotting analysis for SIX1 and FLI1 in (A) A673 SCR and SIX1 KD cells and (B) EWS-502 SCR and SIX1 KD cells. qRT-PCR analysis for (C) EWS/FLI1 expression and (D) SIX1 expression in A673 A1C (dox-inducible EWS/FLI1 KD) cells treated with the indicated time course of 2µg/ml doxycycline. Mean±SD of technical replicates shown. Statistical analysis performed with a one-way ANOVA with post-hoc Dunnett's multiple comparisons test. (E) Western blotting analysis for FLI1 and SIX1 in time course of doxycycline treatment in A673 A1C cells. (F) Western blotting analysis for SIX1 levels in A673 SCR cell with 48h treatment with 2µg/ml doxycycline as a control for E.

Rebuttal Fig. 17. (Revised Manuscript Fig. 7). EWS/FLI1 regulates SIX1 protein levels but not transcription. Representative qRT-PCR expression of (A) EWS/FLI1 and (B) SIX1 in A673 SCR and shFLI1 KD cells. qRT-PCR expression of (C) EWS/FLI1 and (D) SIX1 in EWS-502 SCR and shFLI1 KD cells. For all qRT-PCR experiments, Mean±SD of technical replicates shown. Statistics performed using one-way ANOVA with post-hoc Dunnett's multiple comparisons test. (E) Western blotting analysis of A673 SCR and shFLI1 KD cells for SIX1 levels. (F) Western blotting analysis of EWS-502 SCR and shFLI1 KD cells for SIX1 levels.

competency? This data should be presented, as should some basic histological analyses.

Response: To address this question, we performed tail vein injections with EWS-502 SCR and shSIX1 KD cells to look for effects on metastatic competency. Of note, EWS-502 cells did not tolerate luciferase tagging *in vitro* or *in vivo*, so we performed this experiment with untagged cells. At 33 days post-injection, the mice were sacrificed and dissected to look for evidence of metastasis. Macroscopic metastases were detectable in the livers (and one supra-hilar mass) of 4/9 SIX1 KD mice, but no macroscopic metastases were observed in the EWS-502 SCR control cell injected mice. H&E analysis of the liver sections confirmed the macroscopic findings. These new data, including histologic images, can be found in **Rebuttal Fig. 1, Revised Manuscript Fig. 3D, E**, in response to the same question from Reviewer #1. These data support our finding that SIX1 inhibits metastatic outgrowth of ES cells using multiple models.

5. Figure 6: It is unusual to find so many integrins upregulated in response to Six1-deficiency. Given that all Six1-regulated alpha integrins partner with beta1 integrin, it seems imperative to also deplete beta1 integrin in Six1-deficient ES cells to gauge its importance in driving metastasis of ES cells.

Response: We agree with the reviewer's comment that it is uncommon to see widespread increases in integrin expression with loss of SIX1. However, our CUT&RUN data in both A673 and EWS-502 cells demonstrate that both SIX1 and EWS-FLI1 bind to the promoters of numerous integrin genes, thus directly regulating them (**Manuscript Figs. 8A-D, Rebuttal Fig. 18, Revised Suppl. Fig. 9**). To continue to address this question from the reviewer, we attempted to stably KD integrin $\beta 1$ in A673 and EWS-502 SCR and SIX1 KD cells (due to the fact that Integrin $\beta 1$ interacts with the alpha integrins) to determine if the increases in integrin expression associated with SIX1 loss are responsible for the pro-metastatic phenotype we observe with SIX1 KD. Unfortunately, despite multiple attempts, SIX1 KD cells could not tolerate the concomitant loss of integrin $\beta 1$ (SCR cells were better able to tolerate integrin $\beta 1$ loss in the A673 system, but still had reduced viability *in vitro*)(data not shown). These data suggest that the SIX1 KD cells are likely selectively dependent on integrin expression, particularly integrin heterodimers containing integrin $\beta 1$, to promote survival. Integrin $\beta 1$ inhibition has been demonstrated to induce apoptosis of cancer cells in a number of additional tumor models^{30,31}. These data precluded us from performing a study with integrin $\beta 1$ inhibition *in vivo*, as they suggest that such inhibition may reduce metastatic outgrowth of SIX1 KD cells primarily by inducing

apoptosis, without directly impacting other pro-metastatic features of these cells. However, given the breadth of literature describing pro-metastatic functions of integrins^{6,7,9-11}, combined with *in vitro* data demonstrating that SIX1 KD-mediated integrin expression is responsible for pro-ECM adhesion and pro-invasive phenotypes of SIX1 KD ES cells (Fig. 6 in the manuscript), we feel that we have enough contextual evidence to *suggest* that SIX1-mediated integrin repression plays a role in the anti-metastatic functions of SIX1 in this context. We have been careful within the manuscript to not state that SIX1 definitely requires integrin repression to repress metastasis, but rather to state that our *in vitro* data suggest integrin repression may be a mechanism by which SIX1 represses metastasis in this context.

6. The “Discussion” section needs to be reworked, as it simply restates the findings of the study as opposed to discussing the significance and impact of the work.

Response: We appreciate this comment, and have rewritten the discussion section completely to emphasize the significance and impact of the work and to add additional commentary on the unique nature of this role for SIX1 in ES, rather than to restate our findings.

Reviewer #3:

The authors provide compelling data that the transcription factor SIX1 suppresses integrin subunit expression in Ewings Sarcoma and it is this that explains the unexpected observation that high SIX1 correlates with a metastasis suppressive phenotype. In contrast in multiple carcinomas SIX1 is associated with promoting metastasis and correlating with poor Overall survival. In the manuscript a parallel to the Ewing Sarcoma fusion protein EWS/FLI1 behaviour is drawn in that over-expression of the fusion protein also is metastasis suppressive.

1. Can authors provide STR sequencing proof that shRNA KD lines used in the paper are derived directly from the parental lines.

Response: We have STR profiled both the A673 and EWS-502 SCR and SIX1 KD lines. EWS-502 cells do not have a published STR profile, and therefore our STR profiles do not match any STR profile in publicly available datasets. Nonetheless, we are able to compare the SCR and SIX1 KD STR profiles for EWS-502 to each other and to the parental STR profile and we were able to confirm they were the same line. Both the A673 SCR and SIX1 KD cells matched the published STR profile for the A673 cell line. This information has been added to the methods section of the manuscript.

2. The authors do not explore why SIX1 is metastasis suppressive in Ewings and metastasis promoting in multiple carcinomas. Their explanation in Ewing's is that SIX1 binds to the promoter of multiple integrin subunit genes and suppresses their transcription. Does that happen in the several carcinomas where SIX1 drives metastasis. If it does then the hypothesis presented here is wrong. If SIX1 does not suppress integrins in carcinomas then explain why. At the end of the results the authors imply that SIX1 may function differently in Ewing's versus carcinomas because of the presence of the fusion protein EWS/FLI1. That is easily testable. Eliminate it in the Ewing's lines and introduce it into the carcinoma lines then re-examine the integrin regulation by SIX1.

Response: We appreciate the reviewer's recommended approaches to explore the differential function of SIX1 in ES as compared to its functions in carcinomas. We explored these questions using several approaches. We first looked at whether we saw similar effects of SIX1 expression on integrin expression in the human breast cancer model MCF7. We compared MCF7 cells with low baseline expression of SIX1 to MCF7 cells with overexpression of SIX1 (MCF7-SIX1) and looked at expression of a panel of integrins by RNA-seq (we did not have expression data for ITGA1 and ITGA4 in this dataset). SIX1 overexpression had a mixed effect on integrin expression, as it significantly increased the expression of ITGA2 while significantly decreasing the expression of ITGA5 and ITGA7 (Rebuttal Fig. 19, Response Fig. 4A). No other integrins had significant changes in expression, suggesting that integrin expression in carcinomas is not consistently regulated by SIX1 in the same manner as observed in ES cells. These data suggest a unique pattern of integrin regulation by SIX1 in ES as compared to breast carcinoma. We posit that SIX1 does not uniformly regulate expression of integrins in carcinoma cells due to the absence of expression of EWS/FLI1, however introduction of EWS/FLI1 expression in carcinomas is unlikely to replicate the findings observed in ES, as the cellular context is uniquely important for the function of EWS/FLI1³². This is evidenced by the fact that expression of EWS/FLI1 is toxic in many primary cultured cell lines³³.

3. If EWS/FLI1 is the regulator of the ability of SIX1 to regulate integrin genes then more effort is needed to determine how the two molecules can do this. Does EWS/FLI1 block part of the SIX1 binding site?

Response: To get more clarity as to how, without a direct interaction, SIX1 and EWS/FLI1 may coordinately repress integrin expression (and potentially regulate additional gene expression genome-wide) we performed CUT&RUN in our EWS-502 system to complement that already performed in the A673 system, and compared localization and motif enrichment in both ES systems. Using both CUT&RUN datasets, we performed motif enrichment analysis on the SIX1 and FLI1 sh peaks to identify consensus binding sites enriched at sites bound by SIX1 or EWS/FLI1. In line with the co-localization data, looking for known motifs enriched at SIX1 peak sites throughout the genome, many of the top hits included ETS transcription factors, and most notably there was significant enrichment for the FLI1 consensus binding site (**See Rebuttal Tables 1-4 in response to Reviewer #1 above, or see complete tables in manuscript as Revised Supp. Tables 1-4**). These data suggest that SIX1 is localizing at ETS TF and FLI1 binding motifs throughout the genome. Additional data we generated also suggests that EWS/FLI1 can in fact regulate the protein expression of the SIX1 protein (**Rebuttal Fig. 16 and 17 in response to Reviewer #2 above and in manuscript as Revised Supp. Fig. 8 and Revised Manuscript Fig. 7**), demonstrating that EWS/FLI1 is modulating both SIX1 protein levels and has overlapping binding profiles.

Rebuttal Fig. 20. (Revised Suppl. Fig. 13). SIX1 and EWS/FLI1 have complex roles in co-regulating integrin gene expression. Representative expression levels of (A) SIX1 and (B) EWS-FLI1 in A673 SCR or SIX1 KD cells with non-targeting siRNA (siNT) or siRNA-mediated EWS/FLI1 KD (siEF) via qRT-PCR. Gene expression levels of (C) ITGA1, (D) ITGA4, (E) ITGB1, (F) ITGA4, and (G) ITGA5 via qRT-PCR for the treatment groups described for A and B. Mean±SD of technical replicates shown. Statistical analysis performed with a one-way ANOVA with post-hoc Šidák's multiple comparisons test.

In order to get a better understanding of the relative activity of SIX1 and EWS/FLI1 on integrin gene loci, we performed siRNA-mediated KD of EWS/FLI1 in A673 SCR and SIX1 KD cells. **As seen in Rebuttal Fig. 20, Revised Supp. Figs 13C-E** at select integrin loci (ITGA1, ITGA6, and ITGB1 shown here), SIX1 and EWS/FLI1 KD each led to de-repression of integrin loci, suggesting that both are needed at these loci to repress these integrins. Intriguingly, at other integrins we tested (ITGA4 and ITGA5), SIX1 KD alone led to de-repression, but there appeared to be an antagonistic effect of EWS/FLI1 KD (**Rebuttal Fig. 20, Revised Suppl. Fig. 13F, G**), suggesting that there is a complex system of coordinated integrin regulation by SIX1 and EWS/FLI1, with SIX1 playing a dominant role in integrin gene repression overall. These data are suggestive of coordinate regulation of integrins by both EWS-FLI1 and SIX1, but given the complex nature of this coordinate regulation, future studies will need to be performed to better dissect the precise mechanisms by which SIX1 and EWS-FLI1 regulate specific targets.

4. Please correctly refer to integrin proteins in the text and do not just use the gene names. Integrins are heterodimers so when describing the ligand binding abilities give the full integrin name ie $\alpha 1\beta 1$, $\alpha 2\beta 1$, $\alpha 4\beta 1$, $\alpha 5\beta 1$ and $\alpha 6\beta 1$.

Response: We appreciate the reviewer calling out our incorrect nomenclature use. The manuscript text has been updated to reflect the correct naming for these integrin heterodimers.

5. There are no data on changes in surface expression of integrins. This is easily solved by doing flow cytometry for the several subunits mentioned. Integrins are not trypsin-sensitive so the data will be clean and should correspond with functional data.

Response: As an alternative to flow cytometric analysis to assay for changes in surface expression of integrins, we refer the reviewer to Figures 5C and 5D in the manuscript (**and Rebuttal Figs. 21C, D below**). In this figure,

membrane-specific proteins were isolated to compare protein expression levels of integrin proteins found exclusively in the cellular membrane. As seen in this figure, integrin proteins localized to the membrane were specifically increased in SIX1 KD cells in both systems.

6. The authors highlight that collagen is a major ligand for $\alpha1\beta1$ and $\alpha2\beta1$. Would the authors suggest a potential mechanism of reduced migration in the transwell assays in the presence of antibodies that blocked these integrins, given no collagen was present in the assay?

Response: The integrin-blocking transwell invasion assays were performed with transwells coated with Matrigel, as opposed to uncoated transwells used for migration assays. Matrigel was originally derived from an extract from the Engelberth-Holm-Swarm mouse tumor, which is a poorly-differentiated chondrosarcoma³⁴. Matrigel is a protein extract rich in ECM proteins, including collagen, laminin, and perlecan³⁴. Thus, the integrin-neutralizing antibodies likely worked in this context by reducing this interaction. However, the reviewer raises an interesting point for the migration assays from earlier in the manuscript (Fig. 2A-D), which were done with uncoated transwells. SIX1 KD led to extensive changes in gene expression associated with actin-cytoskeletal dynamics, which can facilitate cytoskeletal reorganization necessary for amoeboid movement, which is characterized by rapid actin polymerization and depolymerization³⁵ enabling a deformable cell morphology with weak cell-ECM contacts^{36,37}. Additionally, SIX1 KD led to increased integrin activation (**Rebuttal Fig. 21, Revised Figs. 5E, F**), which can promote both ECM-dependent and independent motility³⁸⁻⁴⁰.

References

- 1 Chaturvedi, A., Hoffman, L. M., Welm, A. L., Lessnick, S. L. & Beckerle, M. C. The EWS/FLI Oncogene Drives Changes in Cellular Morphology, Adhesion, and Migration in Ewing Sarcoma. *Genes Cancer* **3**, 102-116, doi:10.1177/1947601912457024 (2012).
- 2 Chaturvedi, A. *et al.* Molecular dissection of the mechanism by which EWS/FLI expression compromises actin cytoskeletal integrity and cell adhesion in Ewing sarcoma. *Mol Biol Cell* **25**, 2695-2709, doi:10.1091/mbc.E14-01-0007 (2014).
- 3 Prieur, A., Tirode, F., Cohen, P. & Delattre, O. EWS/FLI-1 silencing and gene profiling of Ewing cells reveal downstream oncogenic pathways and a crucial role for repression of insulin-like growth factor binding protein 3. *Mol Cell Biol* **24**, 7275-7283, doi:10.1128/MCB.24.16.7275-7283.2004 (2004).
- 4 Riggi, N. *et al.* EWS-FLI1 utilizes divergent chromatin remodeling mechanisms to directly activate or repress enhancer elements in Ewing sarcoma. *Cancer Cell* **26**, 668-681, doi:10.1016/j.ccell.2014.10.004 (2014).
- 5 Oikawa, T. & Yamada, T. Molecular biology of the Ets family of transcription factors. *Gene* **303**, 11-34, doi:10.1016/s0378-1119(02)01156-3 (2003).
- 6 Hamidi, H. & Ivaska, J. Every step of the way: integrins in cancer progression and metastasis. *Nat Rev Cancer* **18**, 533-548, doi:10.1038/s41568-018-0038-z (2018).
- 7 Bendas, G. & Borsig, L. Cancer cell adhesion and metastasis: selectins, integrins, and the inhibitory potential of heparins. *Int J Cell Biol* **2012**, 676731, doi:10.1155/2012/676731 (2012).
- 8 Aksorn, N. & Chanvorachote, P. Integrin as a Molecular Target for Anti-cancer Approaches in Lung Cancer. *Anticancer Res* **39**, 541-548, doi:10.21873/anticancer.13146 (2019).
- 9 Guo, W. & Giancotti, F. G. Integrin signalling during tumour progression. *Nat Rev Mol Cell Biol* **5**, 816-826, doi:10.1038/nrm1490 (2004).
- 10 Desgrosellier, J. S. & Cheresh, D. A. Integrins in cancer: biological implications and therapeutic opportunities. *Nat Rev Cancer* **10**, 9-22, doi:10.1038/nrc2748 (2010).
- 11 Benassi, M. S. *et al.* Adhesion molecules in high-grade soft tissue sarcomas: correlation to clinical outcome. *Eur J Cancer* **34**, 496-502, doi:10.1016/s0959-8049(97)10097-1 (1998).
- 12 Shi, K. *et al.* Clinicopathological and prognostic values of fibronectin and integrin alphavbeta3 expression in primary osteosarcoma. *World J Surg Oncol* **17**, 23, doi:10.1186/s12957-019-1566-z (2019).
- 13 Volchenboum, S. L. *et al.* Gene Expression Profiling of Ewing Sarcoma Tumors Reveals the Prognostic Importance of Tumor-Stromal Interactions: A Report from the Children's Oncology Group. *J Pathol Clin Res* **1**, 83-94, doi:10.1002/cjp2.9 (2015).
- 14 Savola, S. *et al.* High Expression of Complement Component 5 (C5) at Tumor Site Associates with Superior Survival in Ewing's Sarcoma Family of Tumour Patients. *ISRN Oncol* **2011**, 168712, doi:10.5402/2011/168712 (2011).
- 15 Ford, H. L., Kabingu, E. N., Bump, E. A., Mutter, G. L. & Pardee, A. B. Abrogation of the G2 cell cycle checkpoint associated with overexpression of HSIX1: a possible mechanism of breast carcinogenesis. *Proc Natl Acad Sci U S A* **95**, 12608-12613, doi:10.1073/pnas.95.21.12608 (1998).
- 16 Micalizzi, D. S. *et al.* The Six1 homeoprotein induces human mammary carcinoma cells to undergo epithelial-mesenchymal transition and metastasis in mice through increasing TGF-beta signaling. *J Clin Invest* **119**, 2678-2690, doi:10.1172/JCI37815 (2009).
- 17 Xu, H. *et al.* Six1 promotes epithelial-mesenchymal transition and malignant conversion in human papillomavirus type 16-immortalized human keratinocytes. *Carcinogenesis* **35**, 1379-1388, doi:10.1093/carcin/bgu050 (2014).
- 18 Ono, H. *et al.* SIX1 promotes epithelial-mesenchymal transition in colorectal cancer through ZEB1 activation. *Oncogene* **31**, 4923-4934, doi:10.1038/onc.2011.646 (2012).

- 19 Blevins, M. A., Towers, C. G., Patrick, A. N., Zhao, R. & Ford, H. L. The SIX1-EYA transcriptional complex as a therapeutic target in cancer. *Expert Opin Ther Targets* **19**, 213-225, doi:10.1517/14728222.2014.978860 (2015).
- 20 Iwanaga, R. *et al.* Expression of Six1 in luminal breast cancers predicts poor prognosis and promotes increases in tumor initiating cells by activation of extracellular signal-regulated kinase and transforming growth factor-beta signaling pathways. *Breast Cancer Res* **14**, R100, doi:10.1186/bcr3219 (2012).
- 21 Zhou, H. *et al.* Identification of a Small-Molecule Inhibitor That Disrupts the SIX1/EYA2 Complex, EMT, and Metastasis. *Cancer Res* **80**, 2689-2702, doi:10.1158/0008-5472.CAN-20-0435 (2020).
- 22 Sannino, G., Marchetto, A., Kirchner, T. & Grunewald, T. G. P. Epithelial-to-Mesenchymal and Mesenchymal-to-Epithelial Transition in Mesenchymal Tumors: A Paradox in Sarcomas? *Cancer Res* **77**, 4556-4561, doi:10.1158/0008-5472.CAN-17-0032 (2017).
- 23 Jain, S. & Kapoor, G. Chemotherapy in Ewing's sarcoma. *Indian J Orthop* **44**, 369-377, doi:10.4103/0019-5413.69305 (2010).
- 24 Hua, L. *et al.* Inhibition of Six1 promotes apoptosis, suppresses proliferation, and migration of osteosarcoma cells. *Tumour Biol* **35**, 1925-1931, doi:10.1007/s13277-013-1258-1 (2014).
- 25 Behbakht, K. *et al.* Six1 overexpression in ovarian carcinoma causes resistance to TRAIL-mediated apoptosis and is associated with poor survival. *Cancer Res* **67**, 3036-3042, doi:10.1158/0008-5472.CAN-06-3755 (2007).
- 26 Yu, C., Zhang, B., Li, Y. L. & Yu, X. R. SIX1 reduces the expression of PTEN via activating PI3K/AKT signal to promote cell proliferation and tumorigenesis in osteosarcoma. *Biomed Pharmacother* **105**, 10-17, doi:10.1016/j.biopha.2018.04.028 (2018).
- 27 Song, K. A. & Faber, A. C. Epithelial-to-mesenchymal transition and drug resistance: transitioning away from death. *J Thorac Dis* **11**, E82-E85, doi:10.21037/jtd.2019.06.11 (2019).
- 28 Zhou, H., Zhang, L., Vartuli, R. L., Ford, H. L. & Zhao, R. The Eya phosphatase: Its unique role in cancer. *Int J Biochem Cell Biol* **96**, 165-170, doi:10.1016/j.biocel.2017.09.001 (2018).
- 29 Apfelbaum, A. A. *et al.* EWS-FLI1 and HOXD13 control tumor cell plasticity in Ewing sarcoma. *Clin Cancer Res*, doi:10.1158/1078-0432.CCR-22-0384 (2022).
- 30 Park, C. C. *et al.* Beta1 integrin inhibitory antibody induces apoptosis of breast cancer cells, inhibits growth, and distinguishes malignant from normal phenotype in three dimensional cultures and in vivo. *Cancer Res* **66**, 1526-1535, doi:10.1158/0008-5472.CAN-05-3071 (2006).
- 31 Rozzo, C., Chiesa, V., Caridi, G., Pagnan, G. & Ponzoni, M. Induction of apoptosis in human neuroblastoma cells by abrogation of integrin-mediated cell adhesion. *Int J Cancer* **70**, 688-698, doi:10.1002/(sici)1097-0215(19970317)70:6<688::aid-ijc11>3.0.co;2-6 (1997).
- 32 Sole, A. *et al.* Unraveling Ewing Sarcoma Tumorigenesis Originating from Patient-Derived Mesenchymal Stem Cells. *Cancer Res* **81**, 4994-5006, doi:10.1158/0008-5472.CAN-20-3837 (2021).
- 33 Deneen, B. & Denny, C. T. Loss of p16 pathways stabilizes EWS/FLI1 expression and complements EWS/FLI1 mediated transformation. *Oncogene* **20**, 6731-6741, doi:10.1038/sj.onc.1204875 (2001).
- 34 Aisenbrey, E. A. & Murphy, W. L. Synthetic alternatives to Matrigel. *Nat Rev Mater* **5**, 539-551, doi:10.1038/s41578-020-0199-8 (2020).
- 35 Paluch, E., Piel, M., Prost, J., Bornens, M. & Sykes, C. Cortical actomyosin breakage triggers shape oscillations in cells and cell fragments. *Biophys J* **89**, 724-733, doi:10.1529/biophysj.105.060590 (2005).
- 36 Wu, J. S. *et al.* Plasticity of cancer cell invasion: Patterns and mechanisms. *Transl Oncol* **14**, 100899, doi:10.1016/j.tranon.2020.100899 (2021).
- 37 Eichinger, L. *et al.* The genome of the social amoeba *Dictyostelium discoideum*. *Nature* **435**, 43-57, doi:10.1038/nature03481 (2005).

- 38 Carragher, N. O. *et al.* Calpain 2 and Src dependence distinguishes mesenchymal and amoeboid modes of tumour cell invasion: a link to integrin function. *Oncogene* **25**, 5726-5740, doi:10.1038/sj.onc.1209582 (2006).
- 39 Tapial Martinez, P., Lopez Navajas, P. & Lietha, D. FAK Structure and Regulation by Membrane Interactions and Force in Focal Adhesions. *Biomolecules* **10**, doi:10.3390/biom10020179 (2020).
- 40 Mitra, S. K. & Schlaepfer, D. D. Integrin-regulated FAK-Src signaling in normal and cancer cells. *Curr Opin Cell Biol* **18**, 516-523, doi:10.1016/j.ceb.2006.08.011 (2006).

Reviewers' Comments:

Reviewer #2:

Remarks to the Author:

The authors have addressed all of my previous concerns, as well as the vast majority of those raised by the other reviewers. As such, the study has been improved significantly and is recommended for publication in Nature Communications.

Reviewer #3:

Remarks to the Author:

The manuscript provides an interesting narrative on the relationship between SIX1 and ES/FL1 and regulation of metastasis. The manuscript still lacks the biological 'smoking gun' that explains how these molecules regulate metastasis. The regulation of integrins is a key biological observation that offered potential mechanistic insight into how the gene regulation by these factors may influence metastasis.

I was pleased that the authors have shown the increased integrin signalling activity correlating with increased beta1 integrin expression and I accept that I missed that membrane integrin expression and matrigel coating of transwells addressed two of my criticisms.

Unfortunately, while it is acknowledged the authors have made some attempt to address inhibition of

integrins, as presented the relationship between the change in integrin expression and the change in metastatic capacity remains correlative. In fairness to the authors they do point out that they do not explicitly state that the change in integrins is the reason for the biological changes. However the paper as a whole provides only correlative observations on metastasis without a clear biological mechanism for its regulation by SIX1 and ES/FL1 .

While it would be expensive to treat mice with antibodies to the upregulated integrin alpha subunits, generating subunit-deficient ES cells using genetic knockout/down of the key alpha subunits is relatively easy to achieve and if injected into mice would confirm if/if not key integrins were responsible for the enhanced experimental metastases. At least then a mechanism can be attributed.

REVIEWERS' COMMENTS

Reviewer #1 (Remarks to the Author):

The authors sufficiently addressed concerns raised in the first review. Performing the in vivo and CUTnRUN experiments in additional models has strengthened their conclusions. They performed more detailed analyses of the relationship between SIX1 and EWS::FLI1 in terms of expression and activity (including integrin expression with double or single knockdowns). They more fully explored integrin activation through FAK and Src phosphorylation. They modified the text to more carefully word their conclusions and put them in better context. Addressing some minor additional questions would help with clarity:

Did they perform FLI1 CUTNRUN on the SIX1 KD cells? Can they show those tracks in the figure for completeness, instead of just SCR? Fig8 a-d.

Response:

We appreciate the reviewer's attention to detail on the CUT&RUN figure. We did perform CUT&RUN using a FLI antibody on the SIX1 KD samples in both the A673 and EWS-502 systems, and made an active decision not to include the FLI1 tracks in the final manuscript due to inconsistency in the data (please see Figures below). While in one A673 and one EWS-502 experiment we saw loss of FLI1 binding at integrin promoters with SIX1 KD, a second experiment performed in the A673 did not show this same trend (see Rebuttal Figs. 1a-c). Thus, more experiments need to be performed to conclusively determine whether SIX1 loss leads to EWS-FLI1 loss at integrin promoters. For this reason, we prefer not to add these data to the current manuscript.

Rebuttal Figure 1a. A673 CUT&RUN Integrin tracks, Set 1. CUT&RUN tracks for SIX1 and FLI1 for A673 SCR and SIX1 KD cells for SIX1 and FLI1 binding at queried integrins. First set.

ITGA1

ITGA2

ITGA4

ITGA5

ITGA6

ITGB1

Rebuttal Figure 1b. EWS-502 CUT&RUN Integrin tracks. CUT&RUN tracks for SIX1 and FLI1 for EWS-502 SCR and SIX1 KD cells for SIX1 and FLI1 binding at queried integrins.

Rebuttal Figure 1c. A673 CUT&RUN Integrin tracks, Set 2. CUT&RUN tracks for SIX1 and FLI1 for A673 SCR and SIX1 KD cells for SIX1 and FLI1 binding at queried integrins. Second set.

The authors mention that SIX1 binding sites were enriched for ETS GGAA motifs and that this has not been shown before in tumors. Did they also observe enrichment of known SIX1 sites or motifs? Or was the localization almost entirely unexpected? It looks like previously described Six1 motifs are not enriched for GGAA or similar sequences (canonical: TCAXXTT).

As observed in Rebuttal Fig. 2a and 2b below, SIX1 motifs were enriched in the A673 CUT&RUN data (both sets, results for set 2 are included in the manuscript) through both a *de novo* motif enrichment approach (Rebuttal Fig. 2a), and a known motif analysis (Rebuttal Fig. 2b). SIX1 was not as high up on the list as expected, suggesting an irregular binding profile for SIX1 in this context. Interestingly, SIX1 did not fall out as a top enriched motif in the EWS-502 CUT&RUN data (not shown), further supporting an atypical binding profile for SIX1 in ES cells. We are continuing to investigate this phenomenon, to see if we can elucidate the cellular or molecular factors responsible for this novel localization pattern of SIX1 in this context.

* - possible false positive

Rank	Motif	P-value	log P-value	% of Targets	% of Background	STD(Bg STD)	Best Match/Details
1		1e-47	-1.105e+02	6.98%	3.99%	25.9bp (29.4bp)	NFY(CCAAT)/Promoter/Homer(0.891) More Information Similar Motifs Found
2		1e-36	-8.401e+01	45.21%	39.28%	26.0bp (30.9bp)	Elk1(ETS)/Hela-Elk1-ChIP-Seq(GSE31477)/Homer(0.887) More Information Similar Motifs Found
3		1e-33	-7.699e+01	39.94%	34.40%	27.4bp (31.8bp)	YY2/MA0748.2/Jaspar(0.679) More Information Similar Motifs Found
4		1e-32	-7.394e+01	4.19%	2.31%	26.9bp (29.7bp)	YY1(Zf)/Promoter/Homer(0.934) More Information Similar Motifs Found
5		1e-27	-6.422e+01	2.75%	1.37%	26.3bp (31.3bp)	TFEC/MA0871.2/Jaspar(0.895) More Information Similar Motifs Found
6		1e-24	-5.727e+01	0.19%	0.01%	26.4bp (27.4bp)	PRDM4/MA1647.1/Jaspar(0.609) More Information Similar Motifs Found
7		1e-23	-5.493e+01	16.21%	12.86%	26.9bp (29.4bp)	PB0110.1_Bcl6b_2/Jaspar(0.912) More Information Similar Motifs Found
8		1e-21	-5.001e+01	9.87%	7.36%	26.5bp (32.1bp)	Zfp57(Zf)/H1-ZFP57.HA-ChIP-Seq(GSE115387)/Homer(0.716) More Information Similar Motifs Found
9		1e-21	-4.899e+01	23.01%	19.34%	27.4bp (29.6bp)	POL010.1_DCE_S_III/Jaspar(0.665) More Information Similar Motifs Found
10		1e-19	-4.585e+01	0.18%	0.01%	24.3bp (11.4bp)	ZFP42/MA1651.1/Jaspar(0.623) More Information Similar Motifs Found
11		1e-16	-3.885e+01	0.12%	0.00%	28.9bp (0.0bp)	PH0126.1_Obox6/Jaspar(0.604) More Information Similar Motifs Found
12		1e-16	-3.872e+01	0.23%	0.03%	24.7bp (30.4bp)	ZNF143/MA0088.2/Jaspar(0.893) More Information Similar Motifs Found
13		1e-16	-3.818e+01	3.22%	2.00%	25.7bp (31.1bp)	ZBTB33/MA0527.1/Jaspar(0.892) More Information Similar Motifs Found
14		1e-15	-3.505e+01	0.11%	0.00%	23.6bp (0.0bp)	PB0036.1_Irf6_1/Jaspar(0.606) More Information Similar Motifs Found
15		1e-13	-3.011e+01	0.12%	0.01%	26.3bp (36.9bp)	TBX3/MA1566.1/Jaspar(0.590) More Information Similar Motifs Found
16		1e-12	-2.812e+01	0.31%	0.07%	26.6bp (29.7bp)	Twist2/MA0633.1/Jaspar(0.689) More Information Similar Motifs Found
17		1e-12	-2.788e+01	0.13%	0.01%	21.0bp (0.0bp)	NRF1/MA0506.1/Jaspar(0.683) More Information Similar Motifs Found
18		1e-12	-2.769e+01	0.09%	0.00%	32.4bp (0.0bp)	SIX1/MA1118.1/Jaspar(0.730) More Information Similar Motifs Found
19 *		1e-11	-2.716e+01	0.29%	0.06%	27.2bp (24.0bp)	ETV2/MA0762.1/Jaspar(0.733) More Information Similar Motifs Found
20 *		1e-11	-2.700e+01	0.11%	0.01%	24.7bp (10.1bp)	ZNF341(Zf)/EBV-ZNF341-ChIP-Seq(GSE113194)/Homer(0.662) More Information Similar Motifs Found

Rebuttal Figure 2a. A673 CUT&RUN Set 2 *de novo* motif analysis results (Top 20). SIX1 motif highlighted in red box.

51		Nrf2(bZIP)/Lymphoblast-Nrf2-ChIP-Seq(GSE37589)/Homer	1e-6	-1.487e+01	0.0000	38.0	0.34%	51.3
52		c-Myc(bHLH)/LNCAP-cMyc-ChIP-Seq(Unpublished)/Homer	1e-6	-1.449e+01	0.0000	1082.0	9.75%	3215.1
53		NPAS(bHLH)/Liver-NPAS-ChIP-Seq(GSE39860)/Homer	1e-6	-1.445e+01	0.0000	1341.0	12.08%	4054.0
54		bHLHE40(bHLH)/HepG2-BHLHE40-ChIP-Seq(GSE31477)/Homer	1e-6	-1.440e+01	0.0000	540.0	4.86%	1500.7
55		LHX9(Homeobox)/Het116-LHX9.V5-ChIP-Seq(GSE116822)/Homer	1e-6	-1.429e+01	0.0000	466.0	4.20%	1273.7
56		Bach1(bZIP)/K562-Bach1-ChIP-Seq(GSE31477)/Homer	1e-6	-1.426e+01	0.0000	31.0	0.28%	38.9
57		KLF14(ZF)/HEK293-KLF14.GFP-ChIP-Seq(GSE58341)/Homer	1e-5	-1.360e+01	0.0000	5467.0	49.25%	17938.6
58		Ap4(bHLH)/AML-Tfap4-ChIP-Seq(GSE45738)/Homer	1e-5	-1.346e+01	0.0000	1520.0	13.69%	4659.4
59		GFY(?)/Promoter/Homer	1e-5	-1.293e+01	0.0000	175.0	1.58%	417.1
60		AP-1(bZIP)/ThioMac-PU.1-ChIP-Seq(GSE21512)/Homer	1e-5	-1.258e+01	0.0000	326.0	2.94%	866.6
61		Fra2(bZIP)/Striatum-Fra2-ChIP-Seq(GSE43429)/Homer	1e-5	-1.239e+01	0.0000	232.0	2.09%	588.0
62		Six2(Homeobox)/NephronProgenitor-Six2-ChIP-Seq(GSE39837)/Homer	1e-5	-1.231e+01	0.0000	545.0	4.91%	1545.7
63		JunB(bZIP)/DendriticCells-JunB-ChIP-Seq(GSE36099)/Homer	1e-5	-1.190e+01	0.0000	246.0	2.22%	633.4
64		ZFX(ZF)/mES-Zfx-ChIP-Seq(GSE11431)/Homer	1e-4	-1.143e+01	0.0001	2509.0	22.60%	7991.9
65		Tgif1(Homeobox)/mES-Tgif1-ChIP-Seq(GSE55404)/Homer	1e-4	-1.095e+01	0.0001	1699.0	15.31%	5313.1
66		Six1(Homeobox)/Myoblast-Six1-ChIP-Chip(GSE20150)/Homer	1e-4	-1.073e+01	0.0001	142.0	1.28%	339.1
67		OCT/OCT(POU,Homeobox)/NPC-Brm1-ChIP-Seq(GSE35496)/Homer	1e-4	-1.024e+01	0.0002	7.0	0.06%	4.0
68		Fos2(bZIP)/3T3L1-Fos2-ChIP-Seq(GSE56872)/Homer	1e-4	-1.013e+01	0.0003	183.0	1.65%	463.6
69		Foxo1(Forkhead)/RAW-Foxo1-ChIP-Seq(Fan_et_al.)/Homer	1e-4	-1.012e+01	0.0003	1140.0	10.27%	3498.2
70		NF-E2(bZIP)/K562-NFE2-ChIP-Seq(GSE31477)/Homer	1e-4	-9.903e+00	0.0003	41.0	0.37%	71.9
71		Hoxd10(Homeobox)/ChickenMSG-Hoxd10.Flag-ChIP-Seq(GSE86088)/Homer	1e-4	-9.857e+00	0.0003	384.0	3.46%	1078.8
72		BATF(bZIP)/Th17-BATF-ChIP-Seq(GSE39756)/Homer	1e-4	-9.846e+00	0.0003	272.0	2.45%	733.0
73		Hoxa9(Homeobox)/ChickenMSG-Hoxa9.Flag-ChIP-Seq(GSE86088)/Homer	1e-4	-9.822e+00	0.0003	1097.0	9.88%	3366.0
74		X-box(HTH)/NPC-H3K4me1-ChIP-Seq(GSE16256)/Homer	1e-4	-9.710e+00	0.0004	132.0	1.19%	318.5
75		ELF5(ETS)/T47D-ELF5-ChIP-Seq(GSE30407)/Homer	1e-4	-9.662e+00	0.0004	799.0	7.20%	2401.5
76		E2F3(E2F)/MEF-E2F3-ChIP-Seq(GSE71376)/Homer	1e-4	-9.540e+00	0.0004	2443.0	22.01%	7835.4
77		Tgif2(Homeobox)/mES-Tgif2-ChIP-Seq(GSE55404)/Homer	1e-4	-9.307e+00	0.0005	2007.0	18.08%	6386.6

Rebuttal Figure 2b. A673 CUT&RUN Set 1 known motif analysis results (51-77). SIX1 motif highlighted in red box.

Reviewer #2 (Remarks to the Author):

The authors have addressed all of my previous concerns, as well as the vast majority of those raised by the other reviewers. As such, the study has been improved significantly and is recommended for publication in Nature Communications.

We appreciate the reviewer's comments, and we thank them for their feedback which we agree has improved the quality of the manuscript significantly.

Reviewer #3 (Remarks to the Author):

The manuscript provides an interesting narrative on the relationship between SIX1 and ES/FL1 and regulation of metastasis. The manuscript still lacks the biological 'smoking gun' that explains how these molecules regulate metastasis. The regulation of integrins is a key biological observation that offered potential mechanistic insight into how the gene regulation by these factors may influence metastasis.

I was pleased that the authors have shown the increased integrin signalling activity correlating with increased beta1 integrin expression and I accept that I missed that membrane integrin expression and matrigel coating of transwells addressed two of my criticisms.

Unfortunately, while it is acknowledged the authors have made some attempt to address inhibition of integrins, as presented the relationship between the change in integrin expression and the change in metastatic capacity remains correlative. In fairness to the authors they do point out that they do not explicitly state that the change in integrins is the reason for the biological changes. However the paper as a whole provides only correlative observations on metastasis without a clear biological mechanism for its regulation by SIX1 and ES/FL1 .

While it would be expensive to treat mice with antibodies to the upregulated integrin alpha subunits, generating subunit-deficient ES cells using genetic knockout/down of the key alpha subunits is relatively easy to achieve and if injected into mice would confirm if/if not key integrins were responsible for the enhanced experimental metastases. At least then a mechanism can be attributed.

We acknowledge that this experiment would allow for a more definitive link between the increased integrin expression observed with SIX1 KD and the increased metastatic propensity observed with SIX1 KD *in vivo*, however this experiment may pose similar challenges and caveats to what we observed when we attempted Integrin B1 KD. We have further clarified the language surrounding this point to clearly state that changes in integrins may not be solely responsible for the induction of metastasis observed in the SIX1 KD cells. Given the literature surrounding the roles of integrins in promoting metastasis, it is likely this change is contributing to the pro-metastatic effect of SIX1 KD in this context, but we acknowledge that this has not been definitively demonstrated in this paper.